# EMERNERF: EMERGENT SPATIAL-TEMPORAL SCENE DECOMPOSITION VIA SELF-SUPERVISION

**Jiawei Yang**[*,¶]**, Boris Ivanovic**[¶]**, Or Litany**[†,¶]**, Xinshuo Weng**[¶]**, Seung Wook Kim**[¶]**, Boyi Li**[¶]**,
Tong Che**[¶]**, Danfei Xu**[$,¶]**, Sanja Fidler**[§,¶]**, Marco Pavone**[‡,¶]**, Yue Wang**[*,¶]

[*] `{yangjiaw,yue.w}@usc.edu`, University of Southern California

[$] `danfei@gatech.edu`, Georgia Institute of Technology

[§] `fidler@cs.toronto.edu`, University of Toronto

[‡] `pavone@stanford.edu`, Stanford University

[†] `orlitany@gmail.com`, Technion

[¶] `{bivanovic,xweng,seungwookk,boyil,tongc}@nvidia.com`, NVIDIA Research

## ABSTRACT

We present `EmerNeRF`, a simple yet powerful approach for learning spatial-temporal representations of dynamic driving scenes. Grounded in neural fields, `EmerNeRF` simultaneously captures scene geometry, appearance, motion, and semantics via self-bootstrapping. `EmerNeRF` hinges upon two core components: First, it stratifies scenes into static and dynamic fields. This decomposition emerges purely from self-supervision, enabling our model to learn from general, in-the-wild data sources. Second, `EmerNeRF` parameterizes an induced flow field from the dynamic field and uses this flow field to further aggregate multi-frame features, amplifying the rendering precision of dynamic objects. Coupling these three fields (static, dynamic, and flow) enables `EmerNeRF` to represent highly-dynamic scenes self-sufficiently, without relying on ground truth object annotations or pre-trained models for dynamic object segmentation or optical flow estimation. Our method achieves state-of-the-art performance in sensor simulation, significantly outperforming previous methods when reconstructing static (+2.93 PSNR) and dynamic (+3.70 PSNR) scenes. In addition, to bolster `EmerNeRF`'s semantic generalization, we lift 2D visual foundation model features into 4D space-time and address a general positional bias in modern Transformers, significantly boosting 3D perception performance (e.g., 37.50% relative improvement in occupancy prediction accuracy on average). Finally, we construct a diverse and challenging 120-sequence dataset to benchmark neural fields under extreme and highly-dynamic settings. See the project page for code, data, and request pre-trained models: `https://emernerf.github.io`

## 1 INTRODUCTION

Perceiving, representing, and reconstructing dynamic scenes is critical for autonomous agents to understand and interact with their environments. Current approaches predominantly build custom pipelines with components dedicated to identifying and tracking static obstacles and dynamic objects (Yang et al., 2023; Guo et al., 2023). However, such approaches require training each component with a large amount of labeled data and devising complex mechanisms to combine outputs across components. To represent static scenes, approaches leveraging neural radiance fields (NeRFs) (Mildenhall et al., 2021) have witnessed a Cambrian explosion in computer graphics, robotics, and autonomous driving, owing to their strong performance in estimating 3D geometry and appearance (Rematas et al., 2022; Tancik et al., 2022; Wang et al., 2023c; Guo et al., 2023). However, without explicit supervision, NeRFs struggle with dynamic environments filled with fast-moving objects, such as vehicles and pedestrians in urban scenarios. In this work, we tackle this long-standing challenge and develop a *self-supervised* technique for building 4D (space-time) representations of dynamic scenes.

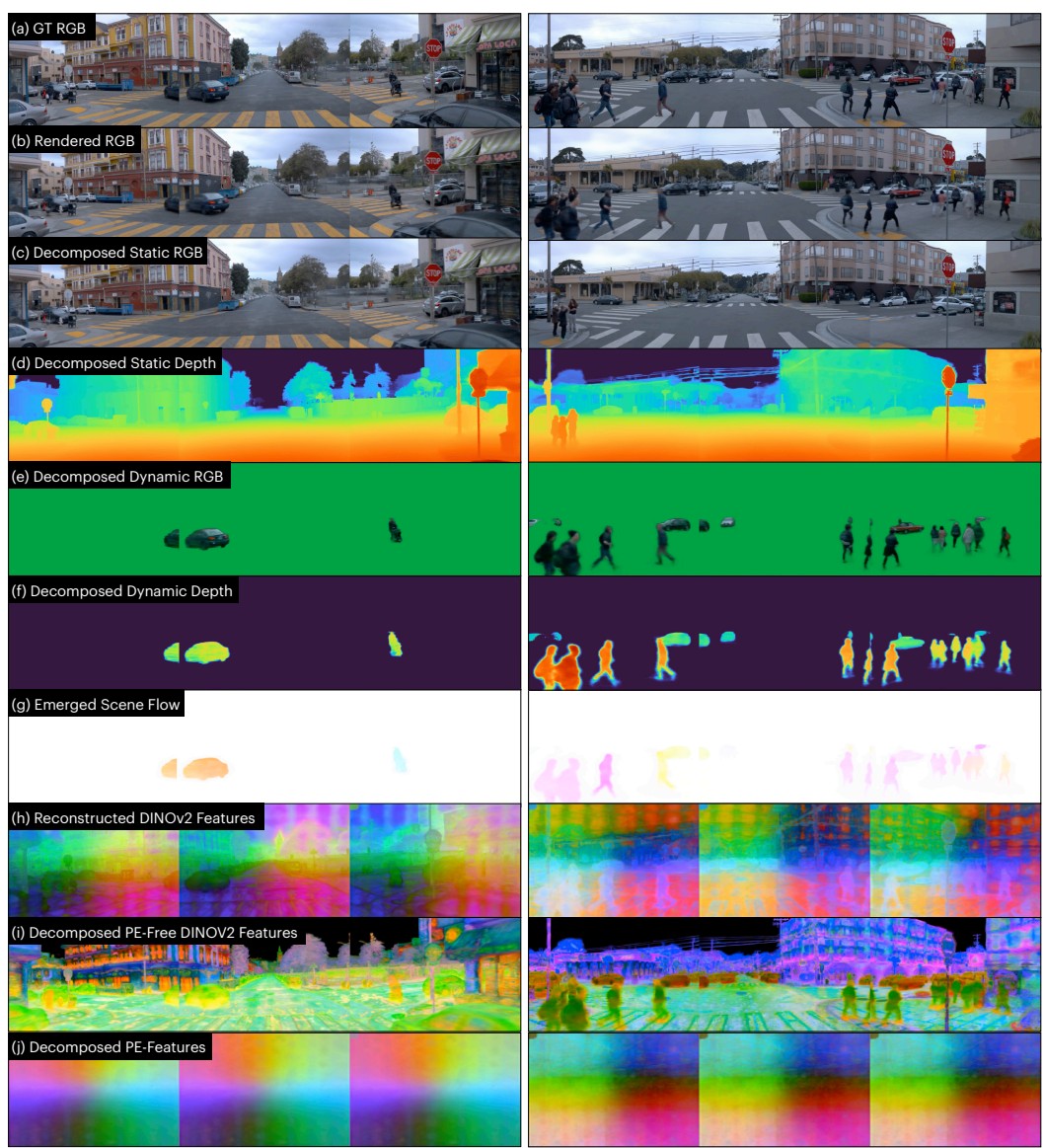

Figure 1: `EmerNeRF` effectively reconstructs photo-realistic dynamic scenes (b), separating them into explicit static (c-d) and dynamic (e-f) elements, all via self-supervision. Notably, (g) scene flows emerge from `EmerNeRF` without any explicit flow supervision. Moreover, `EmerNeRF` can address detrimental positional embedding (PE) patterns observed in vision foundation models (h, j), and lift clean, PE-free features into 4D space (i). Additional visualizations can be found in Appendix C.4.

We consider a common setting where a mobile robot equipped with multiple sensors (e.g., cameras, LiDAR) navigates through a large dynamic environment (e.g., a neighborhood street). The fundamental underlying challenge is to construct an expansive 4D representation with only sparse, transient observations, that is, to reconstruct an entire 4D space-time volume from a *single* traversal of the volume. Unfortunately, this setting violates NeRF's multi-view consistency assumption—each point in the space-time volume will only be observed *once*. Recent works (Yang et al., 2023; Ost et al., 2021) seek to simplify the problem by modeling static scene components (e.g., buildings and trees) and dynamic objects separately, using a combination of neural fields and mesh representations. This decomposition enables exploiting multi-timestep observations to supervise static components, but it often requires costly ground-truth annotations to segment and track dynamic objects. Moreover, despite efforts shown in daily videos (Wang et al., 2021; Gao et al., 2021; Wang et al., 2023b;a; Du et al., 2021), effectively modeling the temporal correspondence of dynamic objects in

autonomous driving scenarios is still largely unexplored. Overall, learning 4D representations of dynamic scenes in the context of autonomous driving remains a formidable challenge.

Towards this end, we present `EmerNeRF`, a self-supervised approach for constructing 4D neural scene representations. As shown in Fig. 1, `EmerNeRF` decouples static and dynamic scene components and estimates 3D scene flows — remarkably, *all from self-supervision*. At a high level, `EmerNeRF` builds a hybrid static-dynamic world representation via a density-regularized objective, generating density for dynamic objects only as necessary (i.e., when points intersect dynamic objects). This representation enables our approach to capture dynamic components and exploit multi-timestep observations to self-supervise static scene elements. To address the lack of cross-observation consistency for dynamic components, we task `EmerNeRF` to predict 3D scene flows and use them to aggregate temporally-displaced features. Intriguingly, `EmerNeRF`'s capability to estimate scene flow emerges naturally from this process, without any explicit flow supervision. Finally, to enhance scene comprehension, we "lift" features from pre-trained 2D visual foundation models (e.g., DINOv1 (Caron et al., 2021), DINOv2 (Oquab et al., 2023)) to 4D space-time. In doing so, we observe and rectify a challenge tied to Transformer-based foundation models: positional embedding (PE) patterns (Fig. 1 (h)). As we will show in §4.3, effectively utilizing such general features greatly improves `EmerNeRF`'s semantic understanding and enables few-shot auto-labeling.

We evaluate `EmerNeRF` on sensor sequences collected by autonomous vehicles (AVs) traversing through diverse urban environments. A critical challenge is that current autonomous driving datasets are heavily imbalanced, containing many simple scenarios with few dynamic objects. To facilitate a focused empirical study and bolster future research on this topic, we present the **N**eRF **O**n-**T**he-**R**oad (NOTR) benchmark, a balanced subsample of 120 driving sequences from the Waymo Open Dataset (Sun et al., 2020) containing diverse visual conditions (lighting, weather, and exposure) and challenging dynamic scenarios. On this benchmark, `EmerNeRF` significantly outperforms previous state-of-the-art NeRF-based approaches (Park et al., 2021b; Wu et al., 2022; Müller et al., 2022; Guo et al., 2023) on scene reconstruction by 2.93 and 3.70 PSNR on static and dynamic scenes, respectively, and by 2.91 PSNR on dynamic novel view synthesis. For scene flow estimation, `EmerNeRF` excels over Li et al. (2021a) by 42.16% in metrics of interest. Additionally, removing PE patterns brings an average improvement of 37.50% relative to using the original, PE pattern-laden features on semantic occupancy prediction. **Contributions.** Our key contributions are fourfold: (1) We introduce `EmerNeRF`, a novel 4D neural scene representation framework that excels in challenging autonomous driving scenarios. `EmerNeRF` performs static-dynamic decomposition and scene flow estimation, all through self-supervision. (2) A streamlined method to tackle the undesired effects of positional embedding patterns from Vision Transformers, which is immediately applicable to other tasks. (3) We introduce the NOTR dataset to assess neural fields in diverse conditions and facilitate future development in the field. (4) `EmerNeRF` achieves state-of-the-art performance in scene reconstruction, novel view synthesis, and scene flow estimation.

## 2 RELATED WORK

**Dynamic scene reconstruction with NeRFs.** Recent works adopt NeRFs (Mildenhall et al., 2021; Müller et al., 2022) to accommodate dynamic scenes (Li et al., 2021b; Park et al., 2021b; Wu et al., 2022; Wang et al., 2021; Gao et al., 2021; Wang et al., 2023a; Du et al., 2021). Earlier methods (Bansal et al., 2020; Li et al., 2022; Wang et al., 2022; Fang et al., 2022) for dynamic view synthesis rely on multiple synchronized videos recorded from different viewpoints, restricting their use for real-world applications in autonomous driving and robotics. Recent methods, such as Nerfies (Park et al., 2021a) and HyperNeRF (Park et al., 2021b), have managed to achieve dynamic view synthesis using a single camera. However, they rely on a strict assumption that all observations can be mapped via deformation back to a canonical reference space, usually constructed from the first timestep.

Of particular relevance to our work are methods like $D^2$NeRF (Wu et al., 2022), SUDS (Turki et al., 2023), and NeuralGroundplans (Sharma et al., 2022). These methods also partition a 4D scene into static and dynamic components. However, $D^2$NeRF underperforms significantly for outdoor scenes due to its sensitivity to hyperparameters and insufficient capacity; NeuralGroundplan relies on synchronized videos from different viewpoints to reason about dynamics; and SUDS, designed for multi-traversal driving logs, largely relies on accurate optical flows derived by pre-trained models and incurs high computational costs due to its expensive flow-based warping losses. In contrast, our

approach can reconstruct an accurate 4D scene representation from a single-traversal log captured by sensors mounted on a self-driving vehicle. Freed from the constraints of pre-trained flow models, `EmerNeRF` exploits and refines its own intrinsic flow predictions, enabling a self-improving loop.

**NeRFs for AV data.** Creating high-fidelity neural simulations from collected driving logs is crucial for the autonomous driving community, as it facilitates the closed-loop training and testing of various algorithms. Beyond SUDS (Turki et al., 2023), there is a growing interest in reconstructing scenes from driving logs. In particular, recent methods excel with static scenes but face challenges with dynamic objects (Guo et al., 2023). While approaches like UniSim (Yang et al., 2023) and NSG (Ost et al., 2021) handle dynamic objects, they depend on ground truth annotations, making them less scalable due to the cost of obtaining such annotations. In contrast, our method achieves high-fidelity simulation results purely through self-supervision, offering a scalable solution.

**Augmenting NeRFs.** NeRF methods are commonly augmented with external model outputs to incorporate additional information. For example, approaches that incorporate scene flow often rely on existing optical flow models for supervision (Li et al., 2021b; Turki et al., 2023; Li et al., 2023b; Wang et al., 2021; Gao et al., 2021; Wang et al., 2023a; Du et al., 2021). They usually require cycle-consistency tests to filter out inconsistent flow estimations; otherwise, the optimization process is prone to failure (Wang et al., 2023b). The Neural Scene Flow Prior (NSFP) (Li et al., 2021a), a state-of-the-art flow estimator, optimizes a neural network to estimate the scene flow at each timestep (minimizing the Chamfer Loss (Fan et al., 2017)). This per-timestep optimization makes NSFP prohibitively expensive. In contrast, our `EmerNeRF` bypasses the need for either pre-trained optical flow models or holistic geometry losses. Instead, our flow field is supervised only by scene reconstruction losses and the flow estimation capability *emerges on its own*. Most recently, 2D signals such as semantic labels or foundation model feature vectors have been distilled into 3D space (Kobayashi et al., 2022; Kerr et al., 2023; Tsagkas et al., 2023; Shafiullah et al., 2022), enabling semantic understanding tasks. In this work, we similarly lift visual foundation model features into 4D space and show their potential for few-shot perception tasks.

## 3 SELF-SUPERVISED SPATIAL-TEMPORAL NEURAL FIELDS

Learning a spatial-temporal representation of a dynamic environment with a multi-sensor robot is challenging due to the sparsity of observations and costs of obtaining ground truth annotations. To this end, our design choices stem from the following key principles: (1) Learn a scene decomposition entirely through self-supervision and avoid using any ground-truth annotations or pre-trained models for dynamic object segmentation or optical flow. (2) Model dynamic element correspondences across time via scene flow. (3) Obtain a mutually reinforcing representation: static-dynamic decomposition and flow estimation can benefit from each other. (4) Improve the semantics of scene representations by leveraging feature lifting and distillation, enabling a range of perception tasks.

Having established several design principles, we are now equipped to describe `EmerNeRF`, a self-supervised approach for efficiently representing both static and dynamic scene components. First, §3.1 details how `EmerNeRF` builds a hybrid world representation with a static and dynamic field. Then, §3.2 explains how `EmerNeRF` leverages an emergent flow field to aggregate temporal features over time, further improving its representation of dynamic components. §3.3 describes the lifting of semantic features from pre-trained 2D models to 4D space-time, enhancing `EmerNeRF`'s scene understanding. Finally, §3.4 discusses the loss function that is minimized during training.

### 3.1 SCENE REPRESENTATIONS

**Scene decomposition.** To enable efficient scene decomposition, we design `EmerNeRF` to be a hybrid spatial-temporal representation. It decomposes a 4D scene into a static field $\mathcal{S}$ and a dynamic field $\mathcal{D}$, both of which are parameterized by learnable hash grids (Müller et al., 2022) $\mathcal{H}_s$ and $\mathcal{H}_d$, respectively. Alternative versatile representations such as Hexplane (Cao & Johnson, 2023) can also be employed. This decoupling offers a flexible and compact 4D scene representation for time-independent features $\mathbf{h}_s = \mathcal{H}_s(\mathbf{x})$ and time-varying features $\mathbf{h}_d = \mathcal{H}_d(\mathbf{x}, t)$, where $\mathbf{x} = (x, y, z)$ is the 3D location of a query point and $t$ denotes its timestep. These features are further transformed into $\mathbf{g}_s$ and $\mathbf{g}_d$ by lightweight MLPs ($g_s$ and $g_d$) and used to predict per-point density $\sigma_s$ and $\sigma_d$:

$$\mathbf{g}_s, \sigma_s = g_s(\mathcal{H}_s(\mathbf{x})) \qquad\qquad \mathbf{g}_d, \sigma_d = g_d(\mathcal{H}_d(\mathbf{x}, t)) \qquad (1)$$

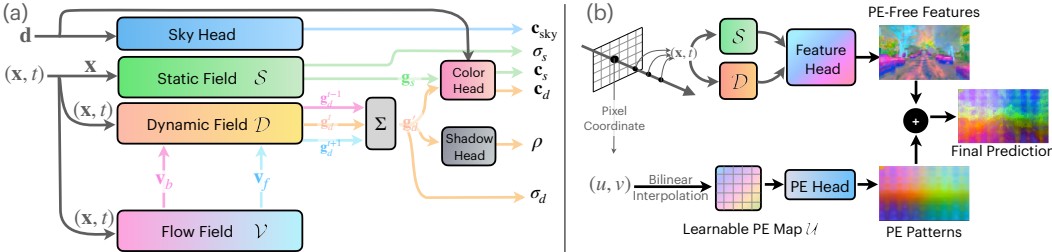

Figure 2: **EmerNeRF Overview.** (a) EmerNeRF consists of a static, dynamic, and flow field $(\mathcal{S}, \mathcal{D}, \mathcal{V})$. These fields take as input either a spatial query $\mathbf{x}$ or spatial-temporal query $(\mathbf{x}, t)$ to generate a static (feature $\mathbf{g}_s$, density $\sigma_s$) pair or a dynamic (feature $\mathbf{g}'_d$, density $\sigma_d$) pair. Of note, we use the forward and backward flows ($\mathbf{v}_f$ and $\mathbf{v}_b$) to generate temporally-aggregated features $\mathbf{g}'_d$ from nearby temporal features $\mathbf{g}_d^{t-1}$, $\mathbf{g}_d^t$, and $\mathbf{g}_d^{t+1}$ (a slight abuse of notation w.r.t. Eq. (8)). These features (along with the view direction $\mathbf{d}$) are consumed by the shared color head which independently predicts the static and dynamic colors $\mathbf{c}_s$ and $\mathbf{c}_d$. (b) EmerNeRF removes undesired positional encoding patterns in ViTs via a learnable PE map followed by a PE head.

**Multi-head prediction.** EmerNeRF uses separate heads for color, sky, and shadow predictions. To maximize the use of dense self-supervision from the static branch, the static and dynamic branches share the same color head $\mathrm{MLP}_{\mathrm{color}}$. This color head takes $(\mathbf{g}_s, \mathbf{d})$ and $(\mathbf{g}_d, \mathbf{d})$ as input, and outputs per-point color $\mathbf{c}_s$ and $\mathbf{c}_d$ for static and dynamic items, where $\mathbf{d}$ is the normalized view direction. Since the depth of the sky is ill-defined, we follow Rematas et al. (2022) and use a separate sky head to predict the sky's color from the frequency-embedded view direction $\gamma(\mathbf{d})$ and features from the static field, where $\gamma(\cdot)$ is a frequency-based positional embedding function, as in Mildenhall et al. (2021). As in Wu et al. (2022), we use a shadow head $\mathrm{MLP}_{\mathrm{shadow}}$ to depict the shadows of dynamic objects. It outputs a scalar $\rho \in [0, 1]$ for dynamic objects, modulating the color intensity predicted by the static field. Collectively, we have:

$$\mathbf{c}_s = \mathrm{MLP}_{\mathrm{color}}(\mathbf{g}_s, \gamma(\mathbf{d})) \qquad\qquad \mathbf{c}_d = \mathrm{MLP}_{\mathrm{color}}(\mathbf{g}_d, \gamma(\mathbf{d})) \qquad (2)$$

$$\mathbf{c}_{\mathrm{sky}} = \mathrm{MLP}_{\mathrm{color\_sky}}(\gamma(\mathbf{d})) \qquad\qquad \rho = \mathrm{MLP}_{\mathrm{shadow}}(\mathbf{g}_d) \qquad (3)$$

**Rendering.** To enable highly-efficient rendering, we use density-based weights to combine results from the static field and dynamic field:

$$\mathbf{c} = \frac{\sigma_s}{\sigma_s + \sigma_d} \cdot (1 - \rho) \cdot \mathbf{c}_s + \frac{\sigma_d}{\sigma_s + \sigma_d} \cdot \mathbf{c}_d \qquad (4)$$

To render a pixel, we use $K$ discrete samples $\{\mathbf{x}_1, \ldots, \mathbf{x}_K\}$ along its ray to estimate the integral of color. The final outputs are given by:

$$\hat{C} = \sum_{i=1}^{K} T_i \alpha_i \mathbf{c}_i + \left(1 - \sum_{i=1}^{K} T_i \alpha_i\right) \mathbf{c}_{\mathrm{sky}} \qquad (5)$$

where $T_i = \prod_{j=1}^{i-1}(1 - \alpha_j)$ is the accumulated transmittance and $\alpha_i$ is the piece-wise opacity.

**Dynamic density regularization.** To facilitate static-dynamic decomposition, we leverage the fact that our world is predominantly static. We regularize dynamic density by minimizing the expectation of the dynamic density $\sigma_d$, which prompts the dynamic field to produce density values only as needed:

$$\mathcal{L}_{\sigma_d} = \mathbb{E}(\sigma_d) = \frac{1}{N_r} \sum_r \frac{1}{K} \sum_{i=1}^{K} \sigma_d(r, i) \qquad (6)$$

where $N_r$ is the number of rays per batch, $K$ represents the number of sampling points along a ray, and $\sigma_d(r, i)$ denotes the predicted dynamic density of the $i$-th point along the $r$-th ray.

### 3.2 EMERGENT SCENE FLOW

**Scene flow estimation.** To capture explicit correspondences between dynamic objects and provide a link by which to aggregate temporally-displaced features, we introduce an additional scene flow

field consisting of a hash grid $\mathcal{V} = \mathcal{H}_v(\mathbf{x}, t)$ and a flow predictor $\mathrm{MLP}_v$. This flow field maps a spatial-temporal query point $(\mathbf{x}, t)$ to a flow vector $\mathbf{v} \in \mathbb{R}^3$, which transforms the query point to its position in the next timestep, given by:

$$\mathbf{v} = \mathrm{MLP}_v(\mathcal{H}_v(\mathbf{x}, t)) \qquad\qquad \mathbf{x}' = \mathbf{x} + \mathbf{v} \qquad\qquad (7)$$

In practice, our flow field predicts both a forward flow $\mathbf{v}_f$ and a backward flow $\mathbf{v}_b$, resulting in a 6-dimensional flow vector for each point.

**Multi-frame feature integration.** Next, we use the link provided by the predicted scene flow to integrate features from nearby timesteps, using a simple weighted summation:

$$\mathbf{g}_d' = 0.25 \cdot g_d(\mathcal{H}_d(\mathbf{x} + \mathbf{v}_b, t-1)) + 0.5 \cdot g_d(\mathcal{H}_d(\mathbf{x}, t)) + 0.25 \cdot g_d(\mathcal{H}_d(\mathbf{x} + \mathbf{v}_f, t+1)) \qquad (8)$$

If not otherwise specified, $\mathbf{g}_d'$ is used by default when the flow field is enabled (instead of $\mathbf{g}_d$ in Eqs. (2) and (3)). This feature aggregation module achieves three goals: 1) It connects the flow field to scene reconstruction losses (e.g., RGB loss) for supervision, 2) it consolidates features, denoising temporal attributes for accurate predictions, and 3) each point is enriched through the shared gradient of its temporally-linked features, enhancing the quality of individual points via shared knowledge.

**Emergent abilities.** We do not use any explicit flow supervision to guide `EmerNeRF`'s flow estimation process. Instead, this capability *emerges* from our temporal aggregation step while optimizing scene reconstruction losses (§3.4). Our hypothesis is that only temporally-consistent features benefit from multi-frame feature integration, and this integration indirectly drives the scene flow field toward optimal solutions — predicting correct flows for all points. Our subsequent ablation studies in Appendix C.1 confirm this: when the temporal aggregation is disabled or gradients of these nearby features are stopped, the flow field fails to learn meaningful results.

## 3.3 VISION TRANSFORMER FEATURE LIFTING

While NeRFs excel at generating high-fidelity color and density fields, they lack in conveying semantic content, constraining their utility for semantic scene comprehension. To bridge this gap, we lift 2D foundation model features to 4D, enabling crucial autonomous driving perception tasks such as semantic occupancy prediction. Although previous works might suggest a straightforward approach (Kerr et al., 2023; Kobayashi et al., 2022), directly lifting features from state-of-the-art vision transformer (ViT) models reveals additional complexities due to positional embeddings (PEs) in transformer models (Fig. 1 (h-j)). In the following sections, we detail how we enhance `EmerNeRF` with a feature reconstruction head, uncover detrimental PE patterns in transformer models, and subsequently mitigate these issues.

**Feature reconstruction head.** Analogous to the color head, we incorporate a feature head $\mathrm{MLP}_{\mathrm{feat}}$ and a feature sky head $\mathrm{MLP}_{\mathrm{feat\_sky}}$ to predict per-point features $\mathbf{f}$ and sky features $\mathbf{f}_{\mathrm{sky}}$, given by:

$$\mathbf{f}_* = \mathrm{MLP}_{\mathrm{feat}}(\mathbf{g}_*), \ \text{where} \ * \in \{s, d\} \qquad\qquad \mathbf{f}_{\mathrm{sky}} = \mathrm{MLP}_{\mathrm{feat\_sky}}(\gamma(\mathbf{d})). \qquad (9)$$

Similar to the color head, we share the feature head among the static and dynamic branches. Rendering these features similarly follows Eq. (5), given by:

$$\hat{F} = \sum_{i=1}^{K} T_i \alpha_i \mathbf{f}_i + \left(1 - \sum_{i=1}^{K} T_i \alpha_i\right) \mathbf{f}_{\mathrm{sky}} \qquad (10)$$

**Positional embedding patterns.** We observe pronounced and undesired PE patterns when using current state-of-the-art foundation models, notably DINOv2 (Oquab et al., 2023) (Fig. 1 (h)). These patterns remain fixed in images, irrespective of 3D viewpoint changes, breaking 3D multi-view consistency. Our experiments (§4.3) reveal that these patterns not only impair feature synthesis results, but also cause a substantial reduction in 3D perception performance.

**Shared learnable additive prior.** We base our solution on the observation that ViTs extract feature maps image-by-image and these PE patterns appear consistently across all images. This suggests that a single, globally-shared PE feature map might be sufficient to capture this shared information. Accordingly, we assume an additive noise model for the PE patterns; that is, they can be independently subtracted from the original features to obtain PE-free features. With this assumption, we construct a learnable and globally-shared 2D feature map $\mathcal{U}$ to compensate for these patterns. This

process is depicted in Fig. 2 (b). For a target pixel coordinate $(u, v)$, we first volume-render a PE-free feature as in Eq. (10). Then, we bilinearly interpolate $\mathcal{U}$ and decode the interpolated feature using a single-layer $\text{MLP}_{\text{PE}}$ to obtain the PE pattern feature, which is then added to the PE-free feature. Formally:

$$\hat{F} = \underbrace{\sum_{i=1}^{K} T_i \alpha_i \mathbf{f}_i + \left(1 - \sum_{i=1}^{k} T_i \alpha_i\right) \mathbf{f}_{\text{sky}}}_{\text{Volume-rendered PE-free feature}} + \underbrace{\text{MLP}_{\text{PE}}\left(\texttt{interp}\left((u, v), \mathcal{U}\right)\right)}_{\text{PE feature}} \quad (11)$$

The grouped terms render "PE-free" features (Fig. 1 (i)) and "PE" patterns (Fig. 1 (j)), respectively, with their sum producing the overall "PE-containing" features (Fig. 1 (h)).

## 3.4 OPTIMIZATION

**Loss functions.** Our method decouples pixel rays and LiDAR rays to account for sensor asynchronization. For pixel rays, we use an L2 loss for colors $\mathcal{L}_{\text{rgb}}$ (and optional semantic features $\mathcal{L}_{\text{feat}}$), a binary cross entropy loss for sky supervision $\mathcal{L}_{\text{sky}}$, and a shadow sparsity loss $\mathcal{L}_{\text{shadow}}$. For LiDAR rays, we combine an expected depth loss with a line-of-sight loss $\mathcal{L}_{\text{depth}}$, as proposed in Rematas et al. (2022). This line-of-sight loss promotes an unimodal distribution of density weights along a ray, which we find is important for clear static-dynamic decomposition. For dynamic regularization, we use a density-based regularization (Eq. 6) to encourage the dynamic field to produce density values only when absolutely necessary. This dynamic regularization loss is applied to both pixel rays ($\mathcal{L}_{\sigma_d(\text{pixel})}$) and LiDAR rays ($\mathcal{L}_{\sigma_d(\text{LiDAR})}$). Lastly, we regularize the flow field with a cycle consistency loss $\mathcal{L}_{\text{cycle}}$. See Appendix A.1 for details. In summary, we minimize:

$$\mathcal{L} = \underbrace{\mathcal{L}_{\text{rgb}} + \mathcal{L}_{\text{sky}} + \mathcal{L}_{\text{shadow}} + \mathcal{L}_{\sigma_d(\text{pixel})} + \mathcal{L}_{\text{cycle}} + \mathcal{L}_{\text{feat}}}_{\text{for pixel rays}} + \underbrace{\mathcal{L}_{\text{depth}} + \mathcal{L}_{\sigma_d(\text{LiDAR})}}_{\text{for LiDAR rays}} \quad (12)$$

**Implementation details.** All model implementation details can be found in Appendix A.

## 4 EXPERIMENTS

In this section, we benchmark the reconstruction capabilities of `EmerNeRF` against prior methods, focusing on static and dynamic scene reconstruction, novel view synthesis, scene flow estimation, and foundation model feature reconstruction. Further ablation studies and a discussion of `EmerNeRF`'s limitations can be found in Appendices C.1 and C.3, respectively.

**Dataset.** While there exist many public datasets with AV sensor data (Caesar et al., 2020; Sun et al., 2020; Caesar et al., 2021), they are heavily imbalanced, containing many simple scenarios with few to no dynamic objects. To remedy this, we introduce **N**eRF **O**n-**T**he-**R**oad (NOTR), a balanced and diverse benchmark derived from the Waymo Open Dataset (Sun et al., 2020). NOTR features 120 unique, hand-picked driving sequences, split into 32 static (the same split as in StreetSurf (Guo et al., 2023)), 32 dynamic, and 56 diverse scenes across seven challenging conditions: ego-static, high-speed, exposure mismatch, dusk/dawn, gloomy, rainy, and night. We name these splits Static-32, Dynamic-32, and Diverse-56, respectively. This dataset not only offers a consistent benchmark for static and dynamic object reconstruction, it also highlights the challenges of training NeRFs on real-world AV data. Beyond simulation, our benchmark offers 2D bounding boxes for dynamic objects, ground truth 3D scene flow, and 3D semantic occupancy—all crucial for driving perception tasks. Additional details can be found in Appendix B.

## 4.1 RENDERING

**Setup.** To analyze performance across various driving scenarios, we test `EmerNeRF`'s scene reconstruction and novel view synthesis capabilities on different NOTR splits. For scene reconstruction, all samples in a log are used for training. This setup probes the upper bound of each method. For novel view synthesis, we omit every 10th timestep, resulting in 10% novel views for evaluation. Our metrics include peak signal-to-noise ratio (PSNR) and structural similarity index (SSIM). For Dynamic scenes, we further leverage ground truth bounding boxes and velocity data to identify dynamic objects and compute "dynamic-only" metrics; and we benchmark against HyperNeRF (Park

Table 1: **Dynamic and static scene reconstruction performance.**

(a) Dynamic-32 Split

| Methods | Scene Reconstruction | | | | Novel View Synthesis | | | |
| | Full Image | | Dynamic-Only | | Full Image | | Dynamic-Only | |
| | PSNR↑ | SSIM↑ | PSNR↑ | SSIM↑ | PSNR↑ | SSIM↑ | DPSNR↑ | SSIM↑ |
|---|---|---|---|---|---|---|---|---|
| D$^2$NeRF | 24.35 | 0.645 | 21.78 | 0.504 | 24.17 | 0.642 | 21.44 | 0.494 |
| HyperNeRF | 25.17 | 0.688 | 22.93 | 0.569 | 24.71 | 0.682 | 22.43 | 0.554 |
| Ours | **28.87** | **0.814** | **26.19** | **0.736** | **27.62** | **0.792** | **24.18** | **0.670** |

(b) Static-32 Split

| Methods | Static Scene Reconstruction | |
| | PSNR↑ | SSIM↑ |
|---|---|---|
| iNGP | 24.46 | 0.694 |
| StreetSurf* | 26.66 | 0.784 |
| Ours | **29.08** | **0.803** |

Table 2: **Scene flow estimation on the NOTR Dynamic-32 split.**

| Methods | EPE3D $(m)$ ↓ | Acc$_5$(%) ↑ | Acc$_{10}$(%) ↑ | $\theta$ (rad) ↓ |
|---|---|---|---|---|
| NSFP (Li et al., 2021a) | 0.365 | 51.76 | 67.36 | 0.84 |
| Ours | **0.014** | **93.92** | **96.27** | **0.64** |

et al., 2021b) and D$^2$NeRF (Wu et al., 2022), two state-of-the-art methods for modeling dynamic scenes. Due to their prohibitive training cost, we only compare against them in the Dynamic-32 split. On the Static-32 split, we disable our dynamic and flow branches, and compare against StreetSurf (Guo et al., 2023) and iNGP Müller et al. (2022) (as implemented by Guo et al. (2023)). We use the official codebases released by these methods, and adapt them to NOTR. To ensure a fair comparison, we augment all methods with LiDAR depth supervision and sky supervision, and disable our feature field. Further details can be found in Appendix A.2.

**Dynamic scene comparisons.** Table 1 (a) shows that our approach consistently outperforms others on scene reconstruction and novel view synthesis. We refer readers to Appendix C.2 for qualitative comparisons. In them, we can see that HyperNeRF (Park et al., 2021b) and D$^2$NeRF (Wu et al., 2022) tend to produce over-smoothed renderings and struggle with dynamic object representation. In contrast, `EmerNeRF` excels in reconstructing high-fidelity static background and dynamic foreground objects, while preserving high-frequency details (evident from its high SSIM and PSNR values). Despite D$^2$NeRF's intent to separate static and dynamic elements, it struggles in complex driving contexts and produces poor dynamic object segmentation (as shown in Fig. C.6). Our method outperforms them both quantitatively and qualitatively. **Static scene comparisons.** While static scene representation is not our main focus, Table 1 (b) shows that `EmerNeRF` achieves state-of-the-art performance in this task as well, outperforming StreetSuRF (Guo et al., 2023) which was designed for static outdoor scenes. Thus, by modeling static and dynamic components, `EmerNeRF` can accurately reconstruct general driving scenes.

## 4.2    FLOW ESTIMATION

**Setup.** We assess `EmerNeRF` on all frames of the Dynamic-32 split, benchmarking against the prior state-of-the-art, NSFP (Li et al., 2021a). Using the Waymo dataset's ground truth scene flows, we compute metrics consistent with Li et al. (2021a): 3D end-point error (EPE3D), calculated as the mean L2 distance between predictions and ground truth for all points; Acc$_5$, representing the fraction of points with EPE3D less than 5cm or a relative error under 5%; Acc$_{10}$, indicating the fraction of points with EPE3D under 10cm or a relative error below 10%; and $\theta$, the average angle error between predictions and ground truths. When evaluating NSFP (Li et al., 2021a), we use their official implementation and remove ground points (our approach does not require such preprocessing).

**Results.** As shown in Table 2, our approach outperforms NSFP across all metrics, with significant leads in EPE3D, Acc$_5$, and Acc$_{10}$. While NSFP (Li et al., 2021a) employs the Chamfer distance loss Fan et al. (2017) to solve scene flow, `EmerNeRF` achieves significantly better results without any explicit flow supervision. These properties naturally emerge from our temporal aggregation step. Appendix C.1 contains additional ablation studies regarding the emergence of flow estimation.

## 4.3    LEVERAGING FOUNDATION MODEL FEATURES

To investigate the impact of ViT PE patterns on 3D perception and feature synthesis, we instantiate versions of `EmerNeRF` with and without our proposed PE decomposition module.

Table 3: **Few-shot semantic occupancy prediction evaluation.** We investigate the influence of positional embedding (PE) patterns on 4D features by evaluating semantic occupancy prediction performance. We report sample-averaged micro-accuracy and class-averaged macro-accuracy.

| PE removed? | ViT model | Static-32 | | Dynamic-32 | | Diverse-56 | | Average of 3 splits | |
|---|---|---|---|---|---|---|---|---|---|
| | | Micro Acc | Macro Acc | Micro Acc | Macro Acc | Micro Acc | Macro Acc | Micro Acc | Macro Acc |
| No | DINOv1 | 43.12% | 52.71% | 47.51% | 54.46% | 43.19% | 51.11% | 44.60% | 52.76% |
| Yes | DINOv1 | **55.02%** | **57.13%** | **57.65%** | **57.77%** | **54.56%** | **55.13%** | **55.74%** | **56.67%** |
| Relative Improvement | | +27.60% | +8.38% | +21.35% | +6.07% | +26.32% | +7.87% | +24.95% | +7.42% |
| No | DINOv2 | 38.73% | 50.30% | 51.43% | 57.03% | 45.22% | 54.37% | 45.13% | 53.90% |
| Yes | DINOv2 | **63.21%** | **59.41%** | **65.08%** | **60.82%** | **57.86%** | **59.00%** | **62.05%** | **59.74%** |
| Relative Improvement | | +63.22% | +18.11% | +26.53% | +6.65% | +27.95% | +8.51% | +37.50% | +10.84% |

Table 4: **Feature synthesis results.** We report the feature-PNSR values under different settings.

| PE removed? | ViT model | Static-32 | Dynamic-32 | Diverse-56 |
|---|---|---|---|---|
| No | DINOv1 | 23.35 | 23.37 | 23.78 |
| Yes | DINOv1 | 23.57 (+0.23) | 23.52 (+0.15) | 23.92 (+0.14) |
| No | DINOv2 | 21.87 | 22.34 | 22.79 |
| Yes | DINOv2 | 22.70 (+0.83) | 22.80 (+0.45) | 23.21 (+0.42) |

**Setup.** We evaluate `EmerNeRF`'s few-shot perception capabilities using the Occ3D dataset (Tian et al., 2023). Occ3D provides 3D semantic occupancy annotations for the Waymo dataset (Sun et al., 2020) in voxel sizes of 0.4m and 0.1m (we use 0.1m). For each sequence, we annotate every 10th frame with ground truth information, resulting in 10% labeled data. Occupied coordinates are input to pre-trained `EmerNeRF` models to compute feature centroids per class. Features from the remaining 90% of frames are then queried and classified based on their nearest feature centroid. We report both micro (sample-averaged) and macro (class-averaged) classification accuracies. All models are obtained from the scene reconstruction setting, i.e., all views are used for training.

**Results.** Table 3 compares the performance of PE-containing 4D features to their PE-free counterparts. Remarkably, `EmerNeRF` with PE-free DINOv2 (Oquab et al., 2023) features sees a maximum relative improvement of 153% in micro-accuracy and an average increase of 78% over its PE-containing counterpart. Intriguingly, although the DINOv1 (Caron et al., 2021) model might appear visually unaffected (Fig. C.7), our results indicate that directly lifting PE-containing features to 4D space-time is indeed problematic. With our decomposition, PE-free DINOv1 features witness an average relative boost of 40.36% in micro-accuracy. As another illustration of PE patterns' impact, DINOv2 features perform significantly *worse* than DINOv1 features without our PE pattern mitigation. Thus, by eliminating PE patterns, the improved performance of DINOv2 over DINOv1 carries over to 3D perception.

**Feature synthesis results.** Table 4 compares the feature-PSNR of PE-containing and PE-free models, showing marked improvements in feature synthesis quality when using our proposed PE decomposition method, especially for DINOv2 (Oquab et al., 2023). While DINOv1 (Caron et al., 2021) appears to be less influenced by PE patterns, our method unveils their presence, further showing that even seemingly unaffected models can benefit from PE pattern decomposition.

## 5  CONCLUSION

In this work, we present `EmerNeRF`, a simple yet powerful approach for learning 4D neural representations of dynamic scenes. `EmerNeRF` effectively captures scene geometry, appearance, motion, and any additional semantic features by decomposing scenes into static and dynamic fields, learning an induced flow field, and optionally lifting foundation model features to a resulting 4D hash grid representation. `EmerNeRF` additionally removes problematic positional embedding patterns that appear when employing Transformer-based foundation model features. Notably, all of these tasks (save for foundation model feature lifting) are learned in a *self-supervised* fashion, without relying on ground truth object annotations or pre-trained models for dynamic object segmentation or optical flow estimation. When evaluated on NOTR, our carefully-selected subset of 120 challenging driving scenes from the Waymo Open Dataset (Sun et al., 2020), `EmerNeRF` achieves state-of-the-art performance in sensor simulation, significantly outperforming previous methods on both static and dynamic scene reconstruction, novel view synthesis, and scene flow estimation. Exciting areas of future work include further exploring capabilities enabled or significantly improved by harnessing foundation model features: few-shot, zero-shot, and auto-labeling via open-vocabulary detection.

## ETHICS STATEMENT

This work primarily focuses on autonomous driving data representation and reconstruction. Accordingly, we use open datasets captured in public spaces which strive to preserve personal privacy by leveraging state-of-the-art object detection techniques to blur people's faces and vehicle license plates. However, these are instance-level characteristics. What requires more effort to manage (and could potentially lead to greater harm) is maintaining a diversity of neighborhoods, and not only in terms of geography, but also population distribution, architectural diversity, and data collection times (ideally repeated traversals uniformly distributed throughout the day and night, for example). We created the NOTR dataset with diversity in mind, hand-picking scenarios from the Waymo Open Dataset (Sun et al., 2020) to ensure a diversity of neighborhoods and scenario types (e.g., static, dynamic). However, as in the parent Waymo Open Dataset, the NOTR dataset contains primarily urban geographies, collected from only a handful of cities in the USA.

## REPRODUCIBILITY STATEMENT

We present our method in §3, experiments and results in §4, implementation details and ablation studies in Appendix A. We benchmark previous approaches and our proposed method using publicly available data and include details of the derived dataset in Appendix B. Additional visualizations, code, models, and data are available either in the appendix or at `https://emernerf.github.io`.

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

# A  IMPLEMENTATION DETAILS

In this section, we discuss the implementation details of `EmerNeRF`. Our code is publicly available, and the pre-trained models will be released upon request. See `emernerf.github.io` for more details.

## A.1  EMERNERF IMPLEMENTATION DETAILS

### A.1.1  DATA PROCESSING

**Data source.** Our sequences are sourced from the `waymo_open_dataset_scene_flow`[1] version, which augments raw sensor data with point cloud flow annotations. For camera images, we employ three frontal cameras: `FRONT_LEFT`, `FRONT`, and `FRONT_RIGHT`, resizing them to a resolution of $640 \times 960$ for both training and evaluation. Regarding LiDAR point clouds, we exclusively use the first return data (ignoring the second return data). We sidestep the rolling shutter issue in LiDAR sensors for simplicity and leave it for future exploration. Dynamic object masks are derived from 2D ground truth camera bounding boxes, with velocities determined from the given metadata. Only objects exceeding a velocity of 1 m/s are classified as dynamic, filtering out potential sensor and annotation noise. For sky masks, we utilize the Mask2Former-architectured ViT-Adapter-L model pre-trained on ADE20k. Note that, the dynamic object masks and point cloud flows are used for *evaluation only*.

**Foundation model feature extraction.** We employ the officially released checkpoints of DINOv2 Oquab et al. (2023) and DINOv1 (Caron et al., 2021), in conjunction with the feature extractor implementation from Amir et al. (2021). For DINOv1, we utilize the ViT-B/16, resizing images to $640 \times 960$ and modifying the model's stride to 8 to further increase the resolution of extracted feature maps. For DINOv2, we use the ViT-B/14 variant, adjusting image dimensions to $644 \times 966$ and using a stride of 7. Given the vast size of the resultant feature maps, we employ PCA decomposition to reduce the feature dimension from 768 to 64 and normalize these features to the [0, 1] range.

### A.1.2  EMERNERF

**Representations.** To account for the efficiency of the 4d neural volume, we build all our scene representations based on iNGP (Müller et al., 2022). We use the iNGP implementations from `tiny-cuda-nn` (Müller, 2021), and use `nerfacc` toolkit (Li et al., 2023a) for acceleration. Following Barron et al. (2023), our static hash grid adopts a resolution between $2^4$ to $2^{13}$ over 10 levels, with a fixed feature-length of 4 for all hash entries. Each level's feature entries are limited to a size of $2^{20}$. With these settings, our model comprises approximately 30M parameters — 18M fewer than StreetSurf's SDF representation (Guo et al., 2023). Our dynamic hash encoder maintains a similar configuration but with a max hash feature map size of $2^{18}$. We anticipate improved performance with larger hash encoders. All MLPs have a hidden layer width of 64. To address camera exposure variations in the wild, we employ per-image 16-dimensional appearance embeddings for the scene reconstruction task and per-camera 16-dimensional embeddings for the novel view synthesis task. Our method builds upon efficient hybrid representations, InstantNGP (Müller et al., 2022). Parallel efforts in improving reconstruction efficiency are also made by Cao & Johnson (2023); Fridovich-Keil et al. (2023); Chen et al. (2022). We believe their representations are also suitable for `EmerNeRF`'s encoders.

**Positional embedding (PE) patterns.** We use a learnable feature map, denoted as $\mathcal{U}$, with dimensions $80 \times 120 \times 32$ ($H \times W \times C$) to accommodate the positional embedding patterns, as discussed in the main text. To decode the PE pattern for an individual pixel located at $(u, v)$, we first sample a feature vector from $\mathcal{U}$ using `F.grid_sample`. Subsequently, a linear layer decodes this feature vector to produce the final PE features.

**Scene range.** To define the axis-aligned bounding box (AABB) of the scene, we utilize LiDAR points. In practice, we uniformly subsample the LiDAR points by a factor of 4 and find the scene boundaries by computing the 2% and 98% percentiles of the world coordinates within the LiDAR

---

[1]`console.cloud.google.com/storage/browser/waymo_open_dataset_scene_flow`

point cloud. However, the LiDAR sensor typically covers only a 75-meter radius around the vehicle. Consequently, an unrestricted contraction mechanism is useful to ensure better performance. Following the scene contraction method detailed in Reiser et al. (2023), we use a piecewise-projective contraction function to project the points falling outside the determined AABB.

**Multi-level sampling.** In line with findings in Mildenhall et al. (2021); Barron et al. (2021), we observe that leveraging extra proposal networks enhances both rendering quality and geometry estimation. Our framework integrates a two-step proposal sampling process, using two distinct iNGP-based proposal models. In the initial step, 128 samples are drawn using the first proposal model which consists of 8 levels, each having a 1-dim feature. The resolution for this model ranges from $2^4$ to $2^9$, and each level has a maximum hash map capacity of $2^{20}$. For the subsequent sampling phase, 64 samples are taken using the second proposal model. This model boasts a maximum resolution of $2^{11}$, but retains the other parameters of the first model. To counteract the "z-aliasing" issue—particularly prominent in driving sequences with thin structures like traffic signs and light poles, we further incorporate the anti-aliasing proposal loss introduced by Barron et al. (2023) during proposal network optimization. A more thorough discussion on this is available in Barron et al. (2023). Lastly, we do not employ spatial-temporal proposal networks, i.e., we don't parameterize the proposal networks with a temporal dimension. Our current implementation already can capture temporal variations from the final scene fields, and we leave integrating a temporal dimension in proposal models for future exploration. For the final rendering, 64 points are sampled from the scene fields.

### A.1.3 OPTIMIZATION

All components in `EmerNeRF` are trained jointly in an end-to-end manner.

**Loss functions.** As we discussed in §3.4, our total loss function is

$$\mathcal{L} = \underbrace{\mathcal{L}_{\text{rgb}} + \mathcal{L}_{\text{sky}} + \mathcal{L}_{\text{shadow}} + \mathcal{L}_{\sigma_d(\text{pixel})} + \mathcal{L}_{\text{cycle}} + \mathcal{L}_{\text{feat}}}_{\text{for pixel rays}} + \underbrace{\mathcal{L}_{\text{depth}} + \mathcal{L}_{\sigma_d(\text{LiDAR})}}_{\text{for LiDAR rays}} \quad \text{(A1)}$$

With $r$ representing a ray and $N_r$ its total number, the individual loss components are defined as:

1. **RGB loss** ($\mathcal{L}_{\text{rgb}}$): Measures the difference between the predicted color ($\hat{C}(r)$) and the ground truth color ($C(r)$) for each ray.

$$\mathcal{L}_{\text{rgb}} = \frac{1}{N_r} \sum_r ||\hat{C}(r) - C(r)||_2^2 \quad \text{(A2)}$$

2. **Sky loss** ($\mathcal{L}_{\text{sky}}$): Measures the discrepancy between the predicted opacity of rendered rays and the actual sky masks. Specifically, sky regions should exhibit transparency. The binary cross entropy (BCE) loss is employed to evaluate this difference. In the equation, $\hat{O}(r)$ is the accumulated opacity of ray $r$ as in Equation (5). $M(r)$ is the ground truth mask with 1 for the sky region and 0 otherwise.

$$\mathcal{L}_{\text{sky}} = 0.001 \cdot \frac{1}{N_r} \sum_r \text{BCE}\left(\hat{O}(r), 1 - M(r)\right) \quad \text{(A3)}$$

3. **Shadow loss** ($\mathcal{L}_{\text{shadow}}$): Penalizes the accumulated squared shadow ratio, following Wu et al. (2022).

$$\mathcal{L}_{\text{shadow}} = 0.01 \cdot \frac{1}{N_r} \sum_r \left(\sum_{i=1}^K T_i \alpha_i \rho_i^2\right) \quad \text{(A4)}$$

4. **Dynamic regularization** ($\mathcal{L}_{\sigma_d(\text{pixel})}$ and $\mathcal{L}_{\sigma_d(\text{LiDAR})}$): Penalizes the mean dynamic density of all points across all rays. This encourages the dynamic branch to generate density only when necessary.

$$\mathcal{L}_{\sigma_\lceil} = 0.01 \cdot \frac{1}{N_r} \sum_r \frac{1}{K} \sum_{i=1}^K \sigma_d(r, i) \quad \text{(A5)}$$

5. **Cycle consistency regularization** ($\mathcal{L}_{cycle}$): Self-regularizes the scene flow prediction. This loss encourages the congruence between the forward scene flow at time $t$ and its corresponding backward scene flow at time $t + 1$.

$$\mathcal{L}_{\text{cycle}} = \frac{0.01}{2} \mathbb{E} \left[ \left[ \text{sg}(\mathbf{v}_f(\mathbf{x}, t)) + \mathbf{v}'_b(\mathbf{x} + \mathbf{v}_f(\mathbf{x}, t), t + 1) \right]^2 \left[ \text{sg}(\mathbf{v}_b(\mathbf{x}, t)) + \mathbf{v}'_f(\mathbf{x} + \mathbf{v}_b(\mathbf{x}, t), t - 1) \right]^2 \right]$$
(A6)

where $\mathbf{v}_f(\mathbf{x}, t)$ denotes forward scene flow at time $t$, $\mathbf{v}'_b(\mathbf{x} + \mathbf{v}_f(\mathbf{x}, t), t + 1)$ is predicted backward scene flow at the forward-warped position at time $t + 1$, sg means stop-gradient operation, and $\mathbb{E}$ represents the expectation, i.e., averaging over all sample points.

6. **Feature loss** ($\mathcal{L}_{\text{feat}}$): Measures the difference between the predicted semantic feature ($\hat{F}(r)$) and the ground truth semantic feature ($F(r)$) for each ray.

$$\mathcal{L}_{\text{feat}} = 0.5 \cdot \frac{1}{N_r} ||\hat{F}(r) - F(r)||_2^2$$
(A7)

7. **Depth Loss** ($\mathcal{L}_{\text{depth}}$): Combines the expected depth loss and the line-of-sight loss, as described in Rematas et al. (2022). The expected depth loss ensures the depth predicted through the volumetric rendering process aligns with the LiDAR measurement's depth. The line-of-sight loss includes two components: a free-space regularization term that ensures zero density for points before the LiDAR termination points and a near-surface loss promoting density concentration around the termination surface. With a slight notation abuse, we have:

$$\mathcal{L}_{\text{exp\_depth}} = \mathbb{E}_r \left[ ||\hat{Z}(r) - Z(r)||_2^2 \right]$$
(A8)

$$\mathcal{L}_{\text{line-of-sight}} = \mathbb{E}_r \left[ \int_{t_n}^{Z(r) - \epsilon} w(t)^2 dt \right] + \mathbb{E}_r \left[ \int_{Z(r) - \epsilon}^{Z(r) + \epsilon} (w(t) - \mathcal{K}_\epsilon (t - Z(r)))^2 \right]$$
(A9)

$$\mathcal{L}_{\text{depth}} = \mathcal{L}_{\text{exp\_depth}} + 0.1 \cdot \mathcal{L}_{\text{line-of-sight}}$$
(A10)

where $\hat{Z}(r)$ represents rendered depth values and $Z(r)$ stands for the ground truth LiDAR range values. Here, the variable $t$ indicates an offset from the origin towards the ray's direction, differentiating it from the temporal variable $t$ discussed earlier. $w(t)$ specifies the blending weights of a point along the ray. $\mathcal{K}_\epsilon(x) = \mathcal{N}(0, (\epsilon/3)^2)$ represents a kernel integrating to one, where $\mathcal{N}$ is a truncated Gaussian. The parameter $\epsilon$ determines the strictness of the line-of-sight loss. Following the suggestions in Rematas et al. (2022), we linearly decay $\epsilon$ from 6.0 to 2.5 during the whole training process.

**Training.** We train our models for 25k iterations using a batch size of 8196. In static scenarios, we deactivate the dynamic and flow branches. Training durations on a single A100 GPU are as follows: for static scenes, feature-free training requires 33 minutes, while the feature-embedded approach takes 40 minutes. Dynamic scene training, which incorporates the flow field and feature aggregation, extends the durations to 2 hours for feature-free and 2.25 hours for feature-embedded representations. To mitigate excessive regularization when the geometry prediction is not reliable, we enable line-of-sight loss after the initial 2k iterations and subsequently halve its coefficient every 5k iterations.

## A.2 BASLINE IMPLEMENTATIONS

For HyperNeRF (Park et al., 2021b) and $D^2$NeRF (Wu et al., 2022), we adopt their officially released JAX implementations and tailor them to our dataset. We train each model for 100k iterations using a batch size of 4096. Training and evaluation for each model take approximately 4 hours on 4 A100 GPUs for a single scene. To ensure a fair comparison, we enhance both models with an additional sky head and provide them with the same depth and sky supervision as in our model. Nevertheless, since neither HyperNeRF nor $D^2$NeRF inherently supports separate sampling of pixel rays and LiDAR rays, we map LiDAR point clouds to the image plane and apply L2 loss for depth supervision. We determine a scale factor from the AABBs derived from LiDAR data to ensure scenes are encapsulated within their predefined near-far range. For StreetSuRF (Guo et al., 2023), we directly quote the reported performance from their paper, as our experiments are conducted under the same setting.

### A.3 RUN TIME ANALYSIS

In Table A.1, we conduct a runtime analysis. Hybrid methods, such as our proposed `EmerNeRF` and StreetSurf (Guo et al., 2023), can be trained in a relatively short duration of 30 minutes to 2 hours. This efficiency comes at the expense of utilizing explicit grids for encodings. In contrast, MLP-based models like D$^2$NeRF Wu et al. (2022) and HyperNeRF Park et al. (2021b) have fewer parameters but require longer training time. It is important to note that the number of parameters in both implicit and hybrid models does not directly correlate with their performance. For instance, as illustrated in Figure 2 of Müller et al. (2022), an MLP with 438k parameters can achieve better image fitting than a hybrid model with a total of 33.6M parameters (10k from MLP and 33.6M from the grids).

Table A.1: **Run-time analysis.** We include the parameter counts for implicit MLPs and explicit encodings like HashGrids (Müller et al., 2022). Additionally, we provide metrics such as the number of GPU hours required for training, the total training iterations, and the batch sizes used, offering a comprehensive overview of the computational requirements. Specifically, StreetSurf (Guo et al., 2023) employs a near-field hash encoder (32M) and a far-field encoder (16M). Our approach has a static hash encoder (30.55M parameters), with the optional inclusion of a dynamic hash encoder (10.49M parameters) and a flow encoder (9.70M parameters).

| Method | # parameters in MLPs | # parameters in grids | # GPU hours | # Iters. | Seconds / 1k iters | Batch Size |
|---|---|---|---|---|---|---|
| *Static:* | | | | | | |
| StreetSurf | 0.1M | 32M + 16M | 1.26hrs | 12,500 | 362.88s | 8192 |
| Ours | 0.065M | 30.55M | 0.58hrs | 25,000 | 83.52s | 8192 |
| *Dynamic:* | | | | | | |
| D2NeRF | 3.03M | 0 | 9.68hrs | 100,000 | 348.48s | 4096 |
| HyperNeRF | 1.32M | 0 | 7.32hrs | 100,000 | 263.52s | 4096 |
| Ours w/o flow | 0.065M | 30.55M + 10.49M | 0.88 hrs | 25,000 | 126.72s | 8192 |
| Ours w/t flow | 0.065M | 30.55M + 10.49M + 9.70M | 2.03 hrs | 25,000 | 292.32s | 8192 |

## B NERF ON-THE-ROAD (NOTR) DATASET

As neural fields gain more attention in autonomous driving, there is an evident need for a comprehensive dataset that captures diverse on-road driving scenarios for NeRF evaluations. To this end, we introduce **N**eRF **O**n-**T**he-**R**oad (NOTR) dataset, a benchmark derived from the Waymo Open Dataset (Sun et al., 2020). NOTR features 120 unique driving sequences, split into 32 static scenes, 32 dynamic scenes, and 56 scenes across seven challenging conditions: ego-static, high-speed, exposure mismatch, dusk/dawn, gloomy, rainy, and nighttime. Examples are shown in Figure B.1.

Beyond images and point clouds, NOTR provides additional resources pivotal for driving perception tasks: bounding boxes for dynamic objects, ground-truth 3D scene flow, and 3D semantic occupancy. We hope this dataset can promote NeRF research in driving scenarios, extending the applications of NeRFs from mere view synthesis to motion understanding, e.g., 3D flows, and scene comprehension, e.g., semantics.

Regarding scene classifications, our static scenes adhere to the split presented in StreetSuRF (Guo et al., 2023), which contains clean scenes with no moving objects. The dynamic scenes, which are frequently observed in driving logs, are chosen based on lighting conditions to differentiate them from those in the "diverse" category. The Diverse-56 samples may also contain dynamic objects, but they are split primarily based on the ego vehicle's state (e.g., ego-static, high-speed, camera exposure mismatch), weather condition (e.g., rainy, gloomy), and lighting difference (e.g., nighttime, dusk/dawn). We provide the sequence IDs of these scenes in our codebase.

## C ADDITIONAL RESULTS

### C.1 ABLATION STUDIES

**Ablation on flow emergence.** Table C.1 provides ablation studies to understand the impact of other components on scene reconstruction, novel view synthesis, and scene flow estimation. For these

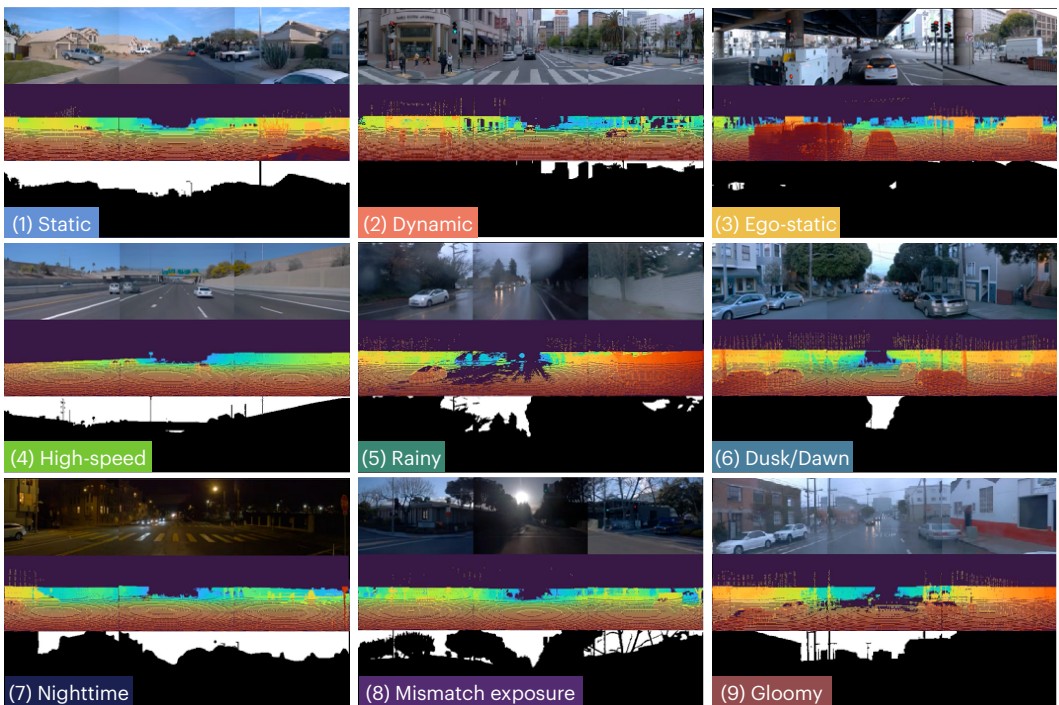

Figure B.1: Samples from the NOTR Benchmark. This comprehensive benchmark contains (1) 32 static scenes, (2) 32 dynamic scenes, and 56 additional scenes across seven categories: (3) ego-static, (4) high-speed, (5) rainy, (6) dusk/dawn, (7) nighttime, (8) mismatched exposure, and (9) gloomy conditions. We include LiDAR visualization in each second row and sky masks in each third row.

Table C.1: **Ablation study**.

| Setting | Scene Reconstruction | | Novel View Synthesis | | Scene Flow estimation |
|---|---|---|---|---|---|
| | Full Image PSNR↑ | Dynamic-Only PSNR↑ | Full Image PSNR↑ | Dynamic-Only PSNR↑ | Flow Acc$_5$(%) ↑ |
| (a) 4D-Only iNGP | 26.55 | 22.30 | 26.02 | 21.03 | - |
| (b) no flow | 26.92 | 23.82 | 26.33 | 23.81 | - |
| (c) no temporal aggregation | 26.95 | 23.90 | 26.60 | 23.98 | 4.53% |
| (d) freeze temporally displaced features before aggregation | 26.93 | 24.02 | 26.78 | 23.81 | 3.87% |
| (e) ours default | **27.21** | **24.41** | **26.93** | **24.07** | **89.74%** |

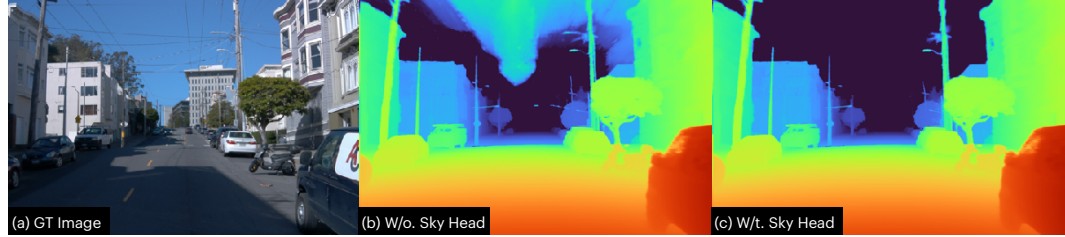

Figure C.1: **Effect of sky head**. The depth of the sky is ill-defined. Using a sky head helps to remove the sky floaters.

Table C.2: Ablation study on different heads.

| Methods | Scene Reconstruction | | | | Novel View Synthesis | | | |
|---|---|---|---|---|---|---|---|---|
| | Full Image | | Dynamic-Only | | Full Image | | Dynamic-Only | |
| | PSNR↑ | SSIM↑ | PSNR↑ | SSIM↑ | PSNR↑ | SSIM↑ | DPSNR↑ | SSIM↑ |
| (a) use separate RGB heads | 28.789 | 0.8042 | 26.107 | 0.7146 | 27.444 | 0.7877 | 24.292 | 0.6656 |
| (b) w/o. sky head | 28.778 | 0.8097 | 26.143 | 0.7255 | 27.533 | 0.7889 | 24.292 | 0.6654 |
| (c) w/t feat. head but w/t PE decomp. | 28.775 | 0.8045 | **26.259** | **0.7280** | **27.844** | **0.7931** | 24.264 | 0.6629 |
| (d) w/t feat. head and w/o PE decomp. | 28.755 | 0.8038 | 26.134 | 0.7238 | 27.469 | 0.7829 | 23.945 | 0.6590 |
| (e) use projected depth | **28.874** | **0.8097** | 26.200 | 0.7275 | 27.532 | 0.7884 | 24.321 | 0.6642 |
| *Ours default: use a shared RGB head for static&dynamic fields, use sky head, w/o feat. head, use LiDAR rays* | | | | | | | | |
| (f) default | 28.873 | 0.8096 | 26.223 | 0.7276 | 27.547 | 0.7885 | **24.323** | **0.6655** |

ablation experiments, all models are trained for 8k iterations, a shorter duration compared to the 25k iterations in the primary experiments. From our observations: (a) Using a full 4D iNGP without the static field results in the worst results, a consequence of the lack of multi-view supervision. (b-e) Introducing hybrid representations consistently improves the results. (c) Omitting the temporal aggregation step or (d) freezing temporal feature gradients (stop the gradients of $\mathbf{g}_d^{t-1}$ and $\mathbf{g}_d^{t+1}$ in Fig. 2) negates the emergence of flow estimation ability, as evidenced in the final column. Combining all these settings yields the best results.

**Ablation on different heads.** Table C.2 provides ablation studies on 5 randomly selected sequences from the Dynamic-32 Split, where we examine the effects of different heads used by our method on rendering performance. These models are trained for 25k iterations. We observe that our method performs reasonably well across all settings. Specifically, (a) using separate RGB heads for static and dynamic fields results in slightly lower rendering quality, particularly for dynamic components, which underscores the effectiveness of our strategy of sharing RGB heads between static and dynamic fields. (b) Removing the sky head does not significantly affect rendering quality, but it does impact depth visualization in sky regions, as shown in Figure C.1. (c, d) Augmenting `EmerNeRF` with semantic feature heads can potentially enhance rendering, but this is contingent upon enabling the proposed positional embedding artifacts decomposition. Without this decomposition, the PSNR scores for dynamic objects decrease from 24.264 (c) to 23.945 (d) in the novel view synthesis task. (f) Utilizing a projected depth map, as opposed to directly sampling LiDAR rays, yields results that are similar to our default setting.

**Ablation on the effect of supervision to flow estimation.** Following the setting in the previous ablation study, we explore how different forms of supervisions and predictions affect the accuracy of flow estimation. The results are shown in Table C.3.

We begin by discussing the impact of depth losses. (b) Using an L2 depth loss results in flow estimations with reasonable scales (i.e., achieving satisfactory $\mathrm{Acc}_5$ and $\mathrm{Acc}_{10}$ results), but these estimates come with a large angle error. (c) Applying a line-of-sight loss yields better results. The line-of-sight loss encourages a concentrated density distribution along a ray towards its termination point—a peaky distribution. This strategy effectively resolves the ambiguity associated with accumulating numerous noisy flows in the 3D scene, as it focuses on the core contributors of these flows: the dynamic objects. (a) Combining these two depth losses provides the optimal performance. Figure C.2 provides a qualitative comparison of methods with and without the line-of-sight loss.

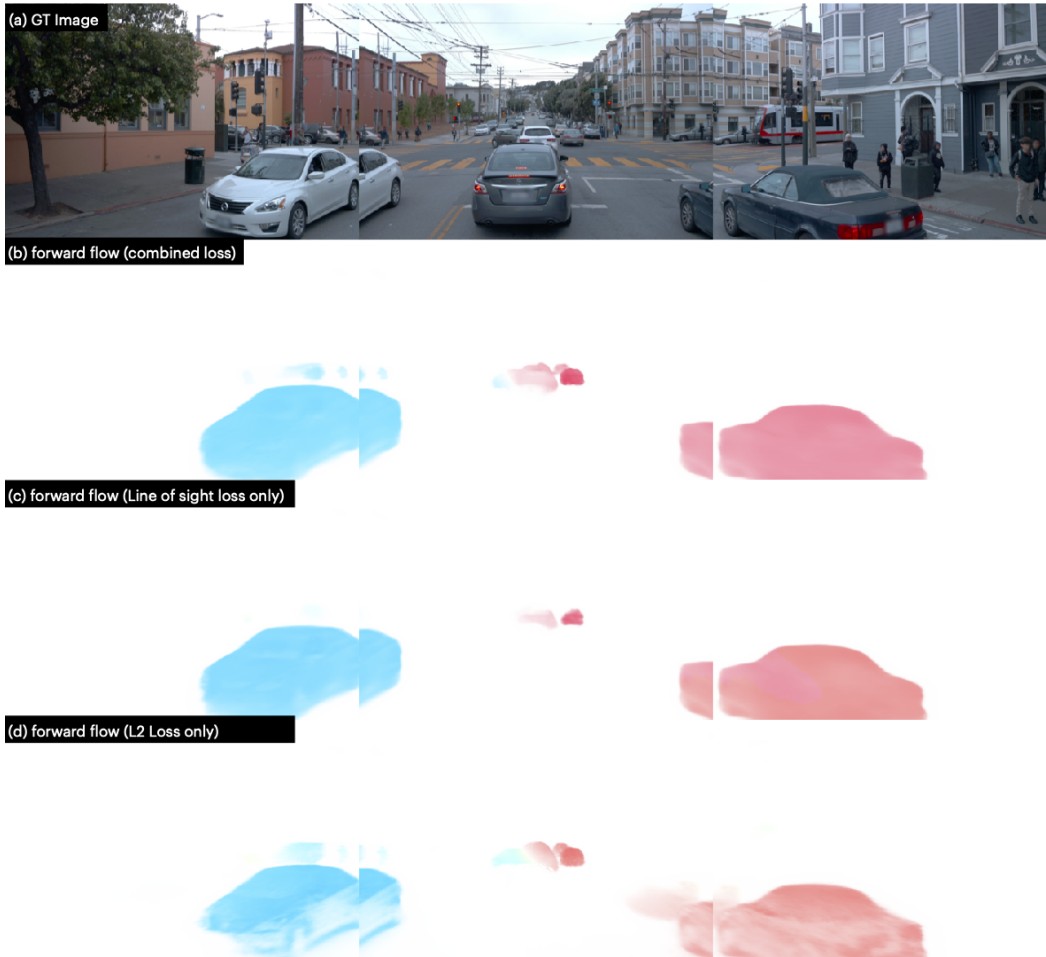

Figure C.2: Qualitative comparative analysis of different depth losses on flow estimation

We next discuss the ablation study on how to utilize flows. In existing NeRF-based methods (Gao et al., 2021; Li et al., 2021b; Wang et al., 2021; 2023a), flow fields are typically employed in a three-step process (referred to as "Color-warping"): 1. **Temporal Warping**: Sampling points are warped to their positions at the previous and/or next timestep to query their corresponding density and colors from nearby frames. 2. **Accumulation**: These densities and colors are then accumulated along corresponding rays in the current time step. Note that for each ray, we perform three individual integrations for density and colors from the past, the current, and the next timestep. This construction results in three distinct per-ray colors. 3. **RGB Loss Computation**: The RGB loss is calculated for each of these three colors.

However, the warped points are occluded in the previous or next timestep, making the RGB loss computation for these points invalid. As a solution, NSFF (Li et al., 2021a) proposed a loss weighting strategy to alleviate this issue. Nonetheless, this approach, referred to as the "warping-based method," is computationally expensive as it requires querying the RGB network three times and rendering three distinct sets of rays.

Our approach differs by aggregating the warped features from both the previous and next timestep. We only query the RGB network with aggregated features and render a single color per ray. This method effectively bypasses the occlusion issues associated with forward- and backward-warped renderings by focusing solely on the current timestep, while leveraging information from nearby frames. Additionally, by decoding the aggregated features through a color MLP, we effectively mitigate potential artifacts introduced by color warping, leading to more robust rendering. To validate this hypothesis, we introduce two variations of our EmerNeRF:

1. Variation 1: Renders forward- and backward-warped rays separately, applying reconstruction losses to each (resulting in three colors per ray in the current timestep), as discussed above. We call this method "Color-warping".
2. Variation 2: Implements color aggregation instead of feature aggregation. This involves predicting forward- and backward-warped per-point colors, aggregating them by taking an average of per-3D location colors from the previous, the current, and the next timestep before taking integration along each ray, and rendering the final output (resulting in one color per ray). We call this method "Color-ensemble". This is to study if the color MLP helps to mitigate feature noise.

Table C.3-(d,e) shows the results. We see that our default method (a) outperforms others (d,e), with the "Color-ensemble" showing the least effective results. The "Color-warping" variant also demonstrated less accurate flow estimations compared to our default method while taking longer training time due to its extra network queryings and color renderings. In addition, all these methods demonstrate promising flow estimation ability. This analysis underscores the potential of utilizing neural fields for flow estimation, especially when depth signals are available, eliminating the need for heuristic methods or pre-trained models to provide optical flow labels. We believe this observation will inspire a broader range of follow-up studies.

Table C.3: **Ablation on the effect of supervision to flow estimation.**

| Settings | EPE3D $(m)\downarrow$ | $Acc_5(\%)\uparrow$ | $Acc_{10}(\%)\uparrow$ | $\theta$ (rad) $\downarrow$ |
|---|---|---|---|---|
| *Combined L2 and line-of-sight loss for depth; feature-ensemble* | | | | |
| (a) Ours default | **0.0130** | 0.9272 | **0.9599** | **0.5175** |
| *Depth supervision:* | | | | |
| (b) L2 loss only | 0.0367 | 0.9077 | 0.9211 | 1.1207 |
| (c) Line-of-sight-loss only | 0.0136 | **0.9296** | 0.9574 | 0.5996 |
| *how to utilize flows:* | | | | |
| (d) Color-warping | 0.0190 | 0.9181 | 0.9370 | 0.6868 |
| (e) Color-ensemble | 0.0414 | 0.8929 | 0.9065 | 1.5708 |

## C.2 QUALITATIVE RESULTS

Figures C.3 and C.4 show qualitative comparisons between our `EmerNeRF` and previous methods under the scene reconstruction setting, while Figure C.5 highlights the enhanced static-dynamic decomposition of our method compared to D$^2$NeRF (Wu et al., 2022). Moreover, Figure C.6 illustrates our method's superiority in novel view synthesis tasks against HyperNeRF (Park et al., 2021b) and D$^2$NeRF (Wu et al., 2022). Our method consistently delivers more realistic and detailed renders.

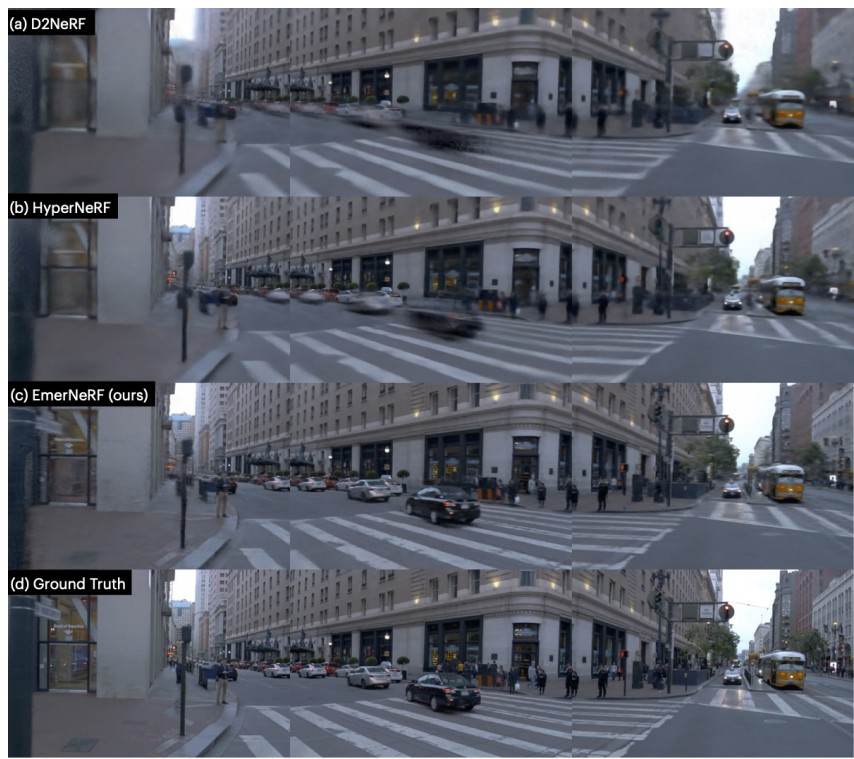

Figure C.3: Qualitative scene reconstruction comparisons.

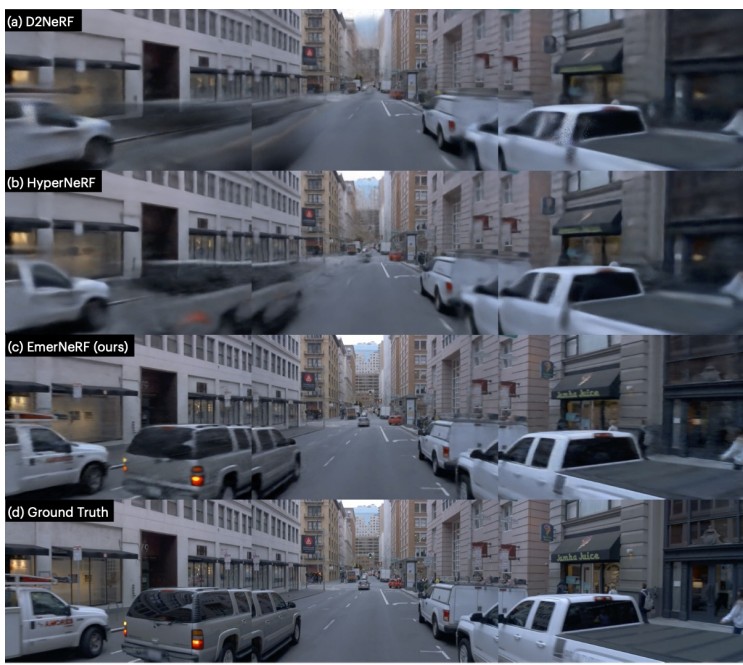

Figure C.4: Qualitative scene reconstruction comparisons.

Notably, HyperNeRF does not decompose static and dynamic components; it provides only composite renders, while our method not only renders high-fidelity temporal views but also precisely separates static and dynamic elements. Furthermore, our method introduces the novel capability of generating dynamic scene flows.

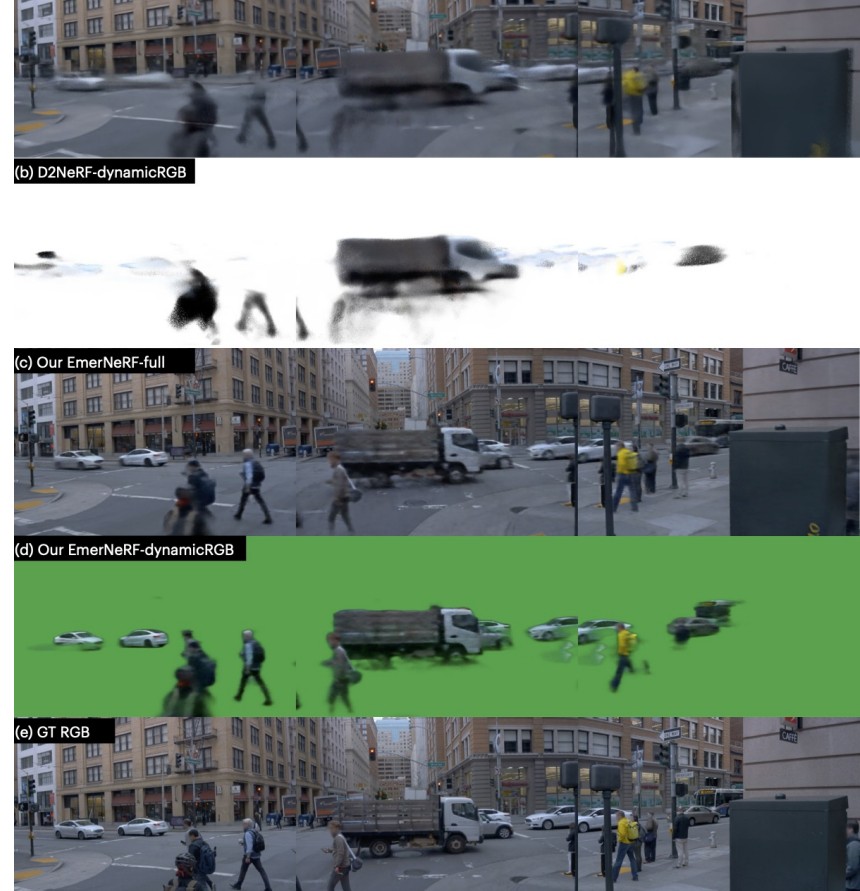

Figure C.5: **Scene decomposition comparisons.** Note that we utilize a green background to blend dynamic objects, whereas D2NeRF's results are presented with a white background.

## C.3   LIMITATIONS

Inline with other methods, `EmerNeRF` does not optimize camera poses and is prone to rolling shutter effects of cameras. Future work to address this issue can investigate joint optimization of pixel-wise camera poses and scene representations. In addition, the balance between geometry and rendering quality remains a trade-off and needs further study. Finally, we leave it for future work to investigate more complicated motion patterns.

## C.4   VISUALIZATIONS

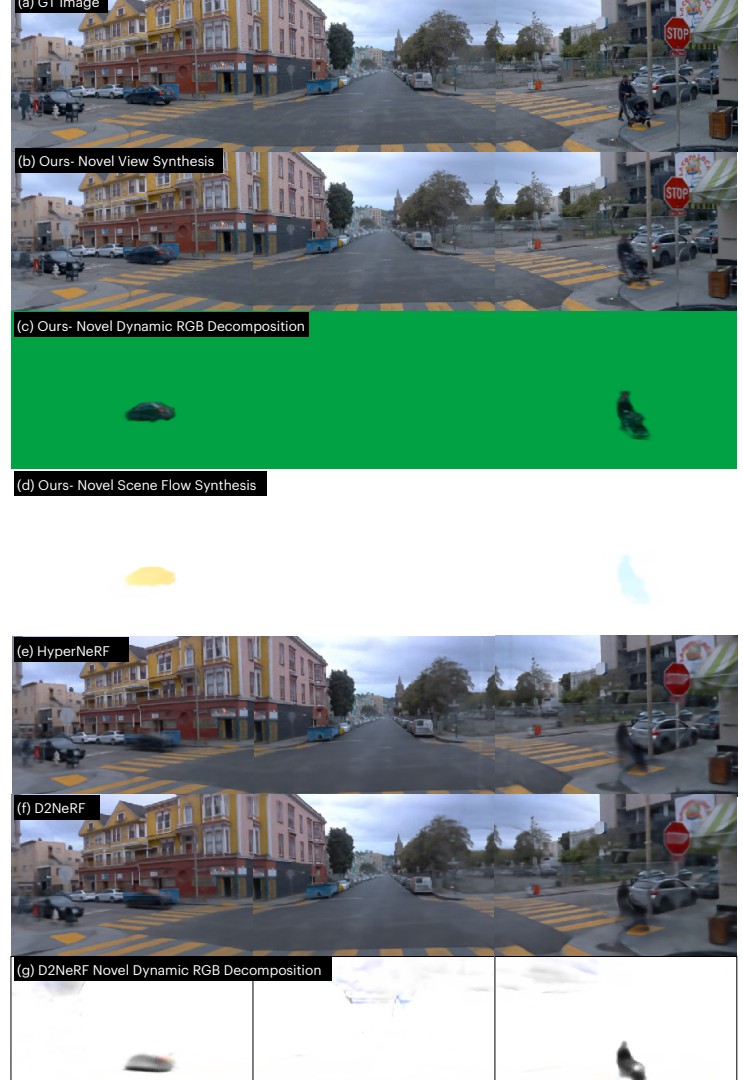

Figure C.6: Qualitative novel temporal view comparison.

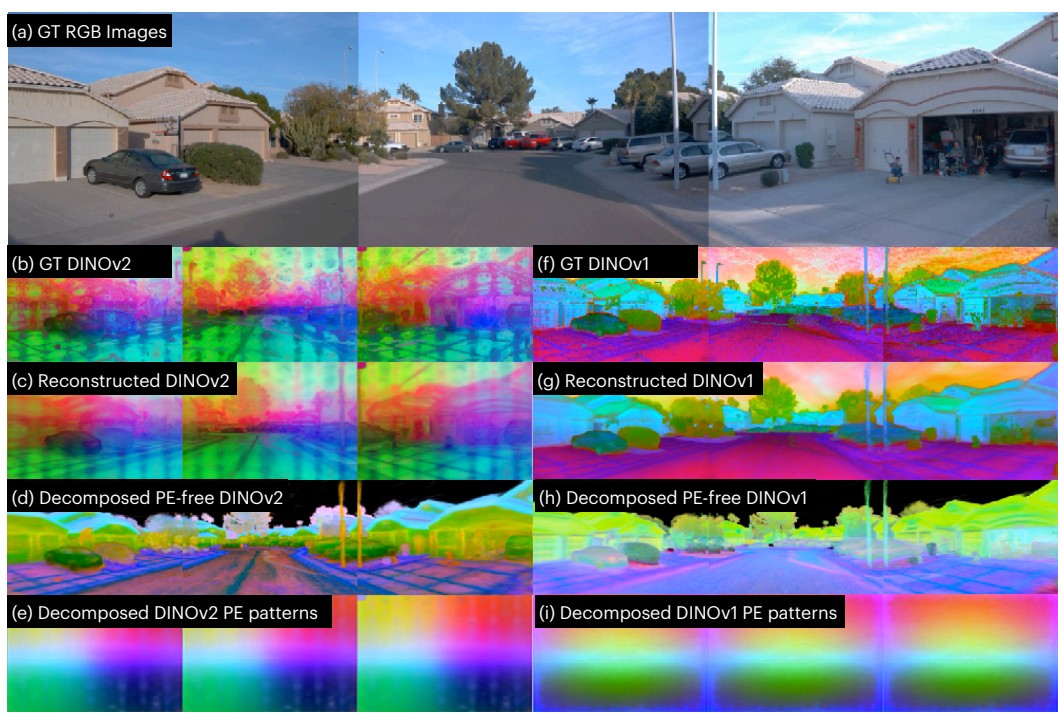

Figure C.7: Different positional embedding patterns in DINOv1 (Caron et al., 2021) and DINOv2 models (Oquab et al., 2023)

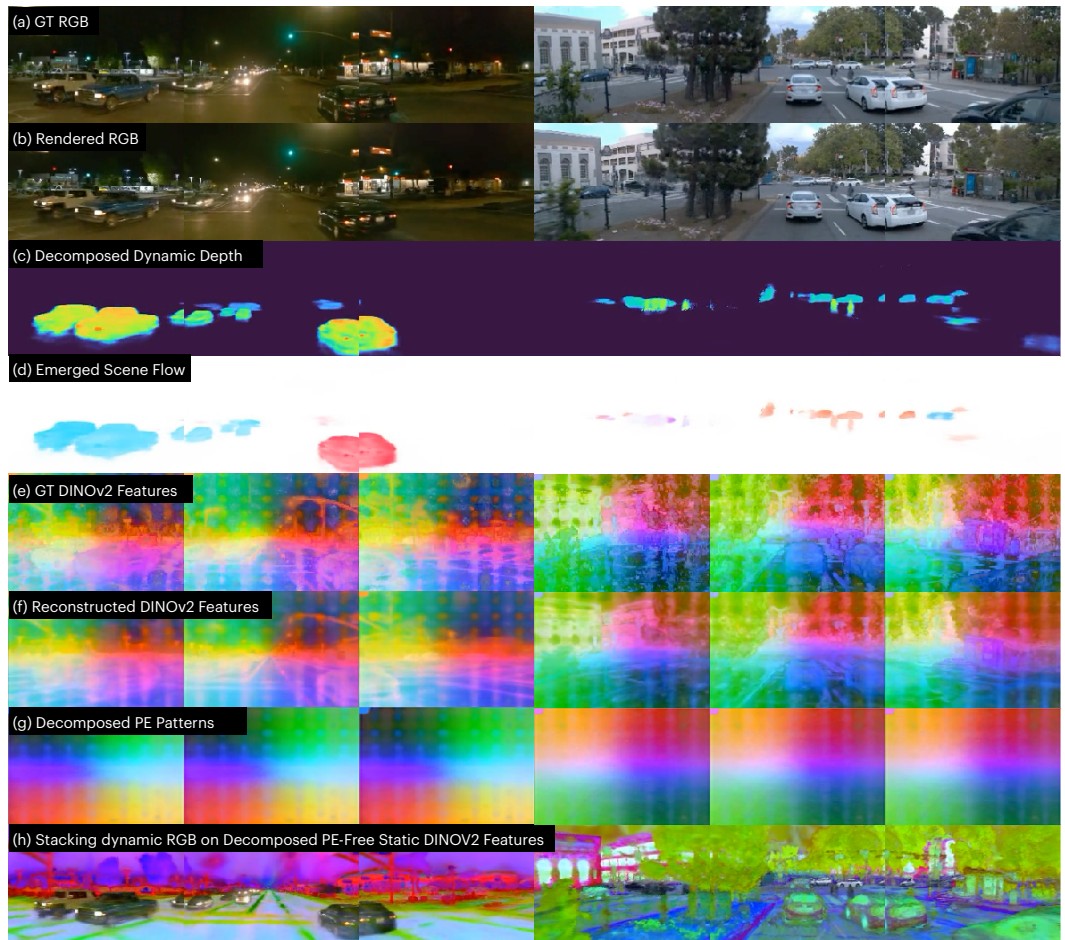

Figure C.8: Scene reconstruction visualizations of `EmerNeRF`. We show (a) GT RGB images, (b) reconstructed RGB images, (c) decomposed dynamic depth, (d) emerged scene flows, (e) GT DINOv2 features, (f) reconstructed DINOv2 features, and (g) decomposed PE patterns. We also stack colors of dynamic objects onto decomposed PE-free static DINOv2 features.

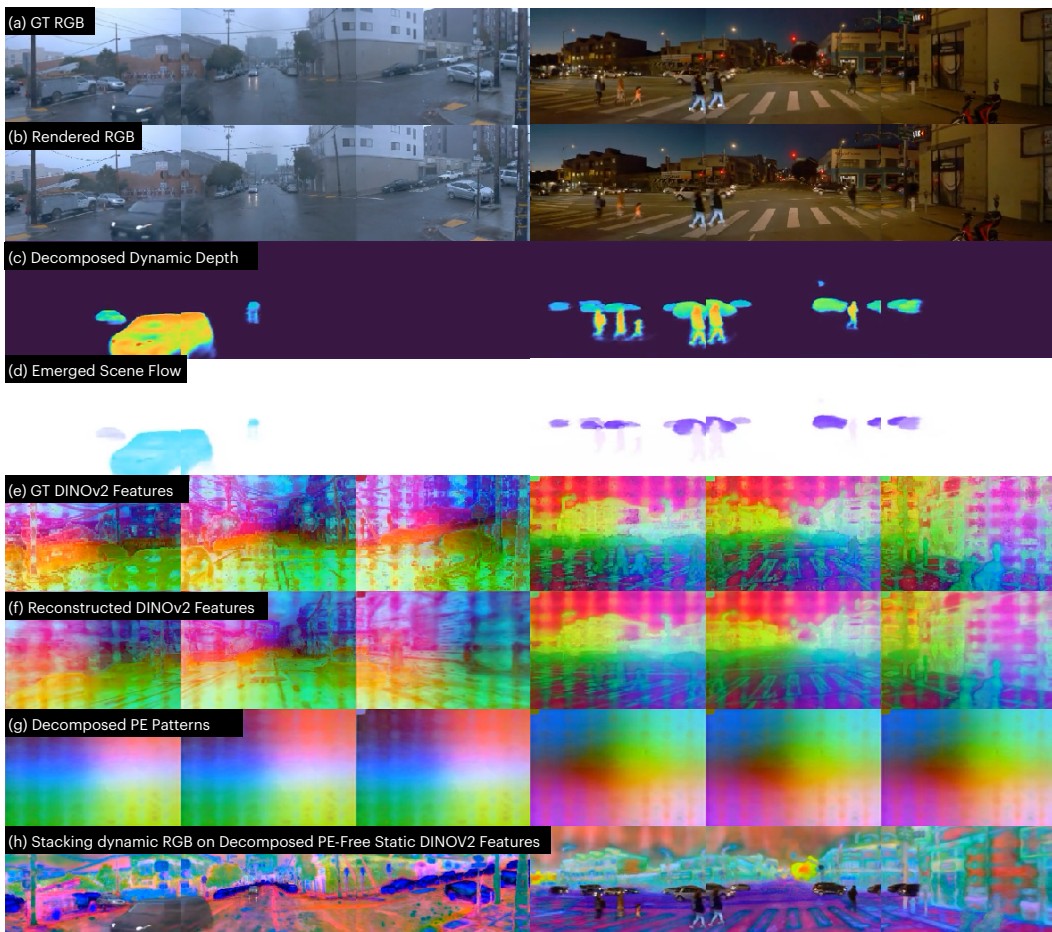

Figure C.9: Scene reconstruction visualizations of EmerNeRF under different lighting conditions. We show (a) GT RGB images, (b) reconstructed RGB images, (c) decomposed dynamic depth, (d) emerged scene flows, (e) GT DINOv2 features, (f) reconstructed DINOv2 features, and (g) decomposed PE patterns. We also stack colors of dynamic objects onto decomposed PE-free static DINOv2 features. EmerNeRF works well under dark environments (left) and discerns challenging scene flows in complex environments (right). Colors indicate scene flows' norms and directions.

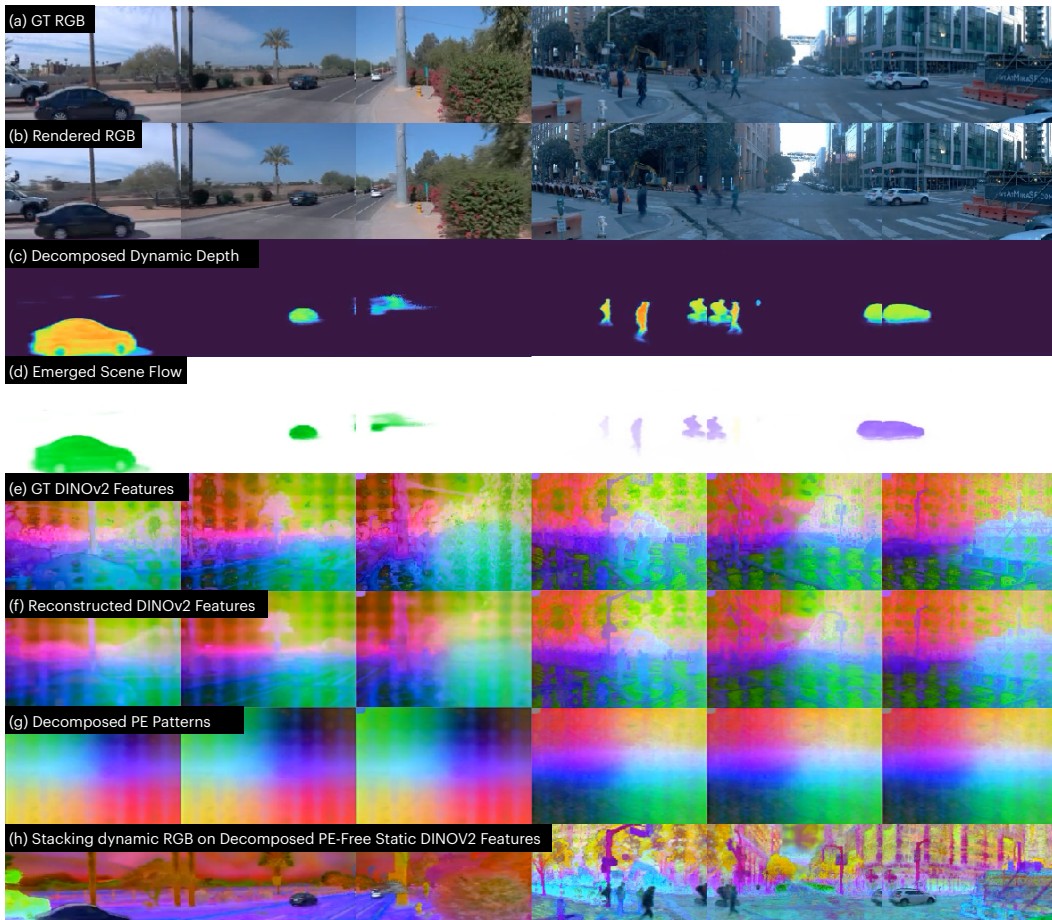

Figure C.10: Scene reconstruction visualizations of `EmerNeRF` under differet lighting and weather conditions. We show (a) GT RGB images, (b) reconstructed RGB images, (c) decomposed dynamic depth, (d) emerged scene flows, (e) GT DINOv2 features, (f) reconstructed DINOv2 features, and (g) decomposed PE patterns. We also stack colors of dynamic objects colors onto decomposed PE-free static DINOv2 features. `EmerNeRF` works well under gloomy environments (left) and discerns fine-grained speed information (right).

