# A    IMPLEMENTATION DETAILS

## A.1    EMERNERF IMPLEMENTATION DETAILS

**Foundation model feature extraction.** We employ the officially released checkpoints of foundation models DINOv2 Oquab et al. (2023) and DINOv1 Caron et al. (2021), in conjunction with the feature extractor implementation from Amir et al. (2021). For DINOv1, we utilize the ViT-B/16, resizing images to 640x960 and modifying the model's stride to 8. For DINOv2, we use the ViT-B/14 variant, adjusting image dimensions to 644x966 and using a stride of 7. Due to the substantial size of their features, we first decrease the feature dimension from 768 to 64 using PCA decomposition. These PCA matrices are computed sequence-by-sequence, with 100k grid feature vectors being randomly sampled from a sequence for computation. Subsequently, we normalize the features to ensure they lie within the [0, 1] range.

**Representations.** To account for the efficiency of the 4d neural volume, we build all our scene representations based on iNGP (Müller et al., 2022). We use the iNGP implementations from `tiny-cuda-nn` (Müller, 2021), and use `nerfacc` toolkit (Li et al., 2023a) for acceleration. Following Barron et al. (2023), our static hash grid adopts a resolution between $2^4$ to $2^{13}$ over 10 levels, with a fixed feature-length of 4 for all hash entries. Each level's feature entries are limited to a size of $2^{20}$. With these settings, our model comprises approximately 30M parameters — 18M fewer than StreetSurf's SDF representation (Guo et al., 2023). Our dynamic hash encoder maintains a similar configuration but with a max hash feature map size of $2^{18}$. We anticipate improved performance with larger hash encoders. All MLPs have a hidden layer width of 64. To address camera exposure variations in the wild, we employ per-image 16-dimensional appearance embeddings for the scene reconstruction task and per-camera 16-dimensional embeddings for the novel view synthesis task.

**Positional embedding (PE) patterns.** We use a feature map size of $80 \times 120 \times 32$ ($H \times W \times C$) to accommodate the positional embedding patterns as discussed in main text. To decode a PE pattern, we first bilinearly upsample this feature map. Subsequently, we employ a linear layer to derive the final PE feature.

**Scene Range.** We rely on LiDAR points to define the scene's axis-aligned bounding box (AABB). Subsequently, we employ the scene contraction method from Reiser et al. (2023) to eliminate spatial points outside the AABB. In practice, we subsample the LiDAR points by a factor of 4 and find the scene boundaries by computing the 2% and 98% percentiles of the world coordinates within the LiDAR point cloud. It's worth noting that the LiDAR sensor typically covers only a 75-meter radius around the vehicle. Consequently, an unrestricted contraction mechanism is useful to ensure better performance.

**Points Sampling.** In line with findings in Mildenhall et al. (2021); Barron et al. (2021), we observe that leveraging extra proposal networks enhances both rendering quality and geometry estimation. Our model encompasses two rounds of proposal sampling: the first with 128 samples and the subsequent with 64 samples. Distinct proposal iNGPs and MLPs are employed for each sampling interation. The first iNGP features 8 levels, each with a 1-dim feature, a resolution spanning 16 to 512, and a maximum hash map capacity per level of $2^{20}$. The second iNGP proposal possesses a peak resolution of 2048, while other parameters are kept consistent. We further incorporate the anti-aliasing proposal loss introduced by Barron et al. (2023) during proposal network optimization. This loss addresses the z-aliasing issue, especially noticeable in driving sequences for slender structures like traffic signs and light poles. For an in-depth discussion, we direct readers to Barron et al. (2023). We do not employ spatial-temporal proposal networks, i.e., we don't parameterize the proposal networks with a temporal dimension. Our current implementation already can capture temporal variations, and we leave integrating a temporal dimension for future exploration. For the final rendering, we sample 64 points from the scene fields.

**Loss functions.** As we discussed in §3.4, our total loss function is

$$\mathcal{L} = \underbrace{\mathcal{L}_{\text{rgb}} + \mathcal{L}_{\text{sky}} + \mathcal{L}_{\text{shadow}} + \mathcal{L}_{\sigma_d(\text{pixel})} + \mathcal{L}_{\text{cycle}} + \mathcal{L}_{\text{feat}}}_{\text{for pixel rays}} + \underbrace{\mathcal{L}_{\text{depth}} + \mathcal{L}_{\sigma_d(\text{LiDAR})}}_{\text{for LiDAR rays}} \tag{A1}$$

In detail, they are:

$$\mathcal{L}_{\text{rgb}} = \frac{1}{N_r}||\hat{C}(r) - C(r)||_2^2 \tag{A2}$$

$$\mathcal{L}_{\text{feat}} = 0.5 \cdot \frac{1}{N_r}||\hat{F}(r) - F(r)||_2^2 \tag{A3}$$

$$\mathcal{L}_{\text{sky}} = 0.001 \cdot \frac{1}{N_r}\text{BCE}\left(\hat{O}(r), M(r)\right) \tag{A4}$$

$$\mathcal{L}_{\text{shadow}} = 0.01 \cdot \frac{1}{N_r}\sum_r\left(\sum_{i=1}^{K}T_i\alpha_i\rho_i^2\right) \tag{A5}$$

$$\mathcal{L}_{\sigma_d} = 0.01 \cdot \mathbb{E}(\sigma_d) \tag{A6}$$

where $r$ denotes a ray, $N_r$ represents the total count of rays, $\hat{C}(r)$ and $C(r)$ denote the predicted and ground truth pixel colors, respectively, and $\hat{F}(r)$ and $F(r)$ are the predicted feature vector and ground truth feature vector. $O(r)$ is the accumulated opacity, given by $\left(1 - \sum_{i=1}^{K}T_i\alpha_i\right)$ as in Equation (5) and BCE stands for binary cross entropy loss. Shadow loss penalizes the accumulated squared shadow ratio, following Wu et al. (2022). The dynamic regularization loss penalizes the mean dynamic density of all points across all rays; we thus use an expectation symbol to avoid notation abuse.

To regularize our predicted scene flow field, we employ a self-regularized cycle consistency loss. This loss encourages the predicted forward scene flow at time $t$ to align with the predicted backward scene flow from the corresponding location at time $t + 1$. Formally, we have:

$$\mathcal{L}_{\text{cycle}} = \frac{1}{2}\mathbb{E}\left[\left[\text{sg}(\mathbf{v}_f(\mathbf{x}, t)) + \mathbf{v}_b'(\mathbf{x} + \mathbf{v}_f(\mathbf{x}, t), t+1)\right]^2 + \left[\text{sg}(\mathbf{v}_b(\mathbf{x}, t)) + \mathbf{v}_f'(\mathbf{x} + \mathbf{v}_b(\mathbf{x}, t), t-1)\right]^2\right] \tag{A7}$$

where $\mathbf{v}_f(\mathbf{x}, t)$ is the forward scene flow predicted at time $t$, $\mathbf{v}_b'(\mathbf{x} + \mathbf{v}_f(\mathbf{x}, t), t+1)$ is the predicted backward scene flow for the forward-warped location at time $t + 1$, i.e., $(\mathbf{x} + \mathbf{v}_f(\mathbf{x}, t), t + 1)$. sg denotes the stop-gradient operation. The $\mathcal{L}_{depth}$ is a combination of expected depth and line-of-sight loss as in Rematas et al. (2022). We refer readers to their work for more details.

**Training.** We train our representations for 25k iterations with a batch size of 8196 unless otherwise specified. For static scenes, we deactivate both the dynamic and flow branches. On a single A100 GPU for a static scene, feature-free training takes 35 minutes, while feature-embedded takes 1.3 hours due to extra computations. Dynamic scenes' training takes longer because of the additional cost of the flow field and the feature aggregation step: on a single A100 GPU, this takes about 1.5hrs and 3h for feature-free and feature-embedded representations. We enable line-of-sight loss after the first 2k iterations to avoid its over-strong regularization. Then, we decay its coefficient by a factor of 0.5 every 5k iterations.

## A.2 BASELINE IMPLEMENTATIONS

For HyperNeRF (Park et al., 2021b) and D$^2$NeRF (Wu et al., 2022), we adopt their officially released JAX implementations and tailor them to our dataset. We train each model for 100k iterations using a batch size of 4096. Training and evaluation for each model take approximately 4 hours on 4 A100 GPUs for a single scene. To ensure a fair comparison, we enhance both models with an additional sky head and provide them with the same depth and sky supervision as in our model. Nevertheless, since neither HyperNeRF nor D$^2$NeRF inherently supports separate sampling of pixel rays and LiDAR rays, we map LiDAR point clouds to the image plane and apply L2 loss for depth supervision. We determine a scale factor from the AABBs derived from LiDAR data to ensure scenes are encapsulated within their predefined near-far range.

For StreetSuRF (Guo et al., 2023), we use their official code. To maintain consistency with our setup, we deactivate their monocular "normal" supervision. Additionally, we noted that StreetSuRF preprocesses LiDAR rays in a manner distinct from our approach. To maintain fairness, we modify their code to accommodate our preprocessed LiDAR rays. Both StreetSuRF and our method employ an error-map-based sampling technique. Specifically, we store RGB error maps at a reduced resolution every 2k iterations, using these maps to determine pixel significance. Rays are then sampled

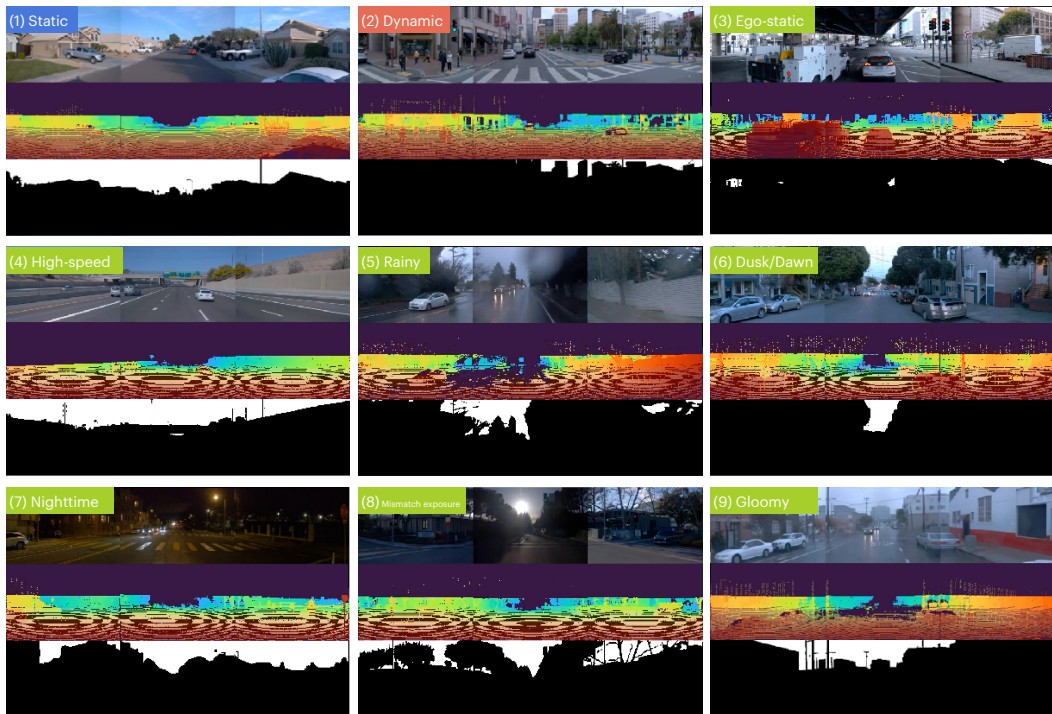

Figure B.1: Samples from the NOTR Benchmark. This comprehensive benchmark contains (1) 32 static scenes, (2) 32 dynamic scenes, and 56 additional scenes across seven categories: (3) ego-static, (4) high-speed, (5) rainy, (6) dusk/dawn, (7) nighttime, (8) mismatched exposure, and (9) gloomy conditions. We include LiDAR visualization in each second row and sky masks in each third row.

based on this pixel significance. The sampling approach yields a slight improvement in scene PSNR by 0.2 for both StreetSuRF and our method.

## B NeRF On-The-Road (NOTR) Dataset

As neural fields gain more attention in autonomous driving, there is an evident need for a comprehensive dataset that captures diverse on-road driving scenarios for NeRF evaluations. To this end, we introduce **N**eRF **O**n-**T**he-**R**oad (NOTR) dataset, a benchmark derived from the Waymo Open Dataset (Sun et al., 2020). NOTR features 120 unique driving sequences, split into 32 static scenes, 32 dynamic scenes, and 56 scenes across seven challenging conditions: ego-static, high-speed, exposure mismatch, dusk/dawn, gloomy, rainy, and nighttime. Examples are shown in Figure B.1.

Beyond images and point clouds, NOTR provides additional resources pivotal for driving perception tasks: bounding boxes for dynamic objects, ground-truth 3D scene flow, and 3D semantic occupancy. We hope this dataset can promote NeRF research in driving scenarios.

Regarding scene classifications, our static scenes adhere to the split presented in StreetSuRF (Guo et al., 2023), which contains clean scenes with no moving objects. The dynamic scenes, which are frequently observed in driving logs, are chosen based on lighting conditions to differentiate them from those in "diverse"" category. The Diverse-56 samples may also contain dynamic objects, but they are splitted primarily based on the ego vehicle's state (e.g., ego-static, high-speed, camera exposure mismatch), weather condition (e.g., rainy, gloomy), and lighting difference (e.g., nighttime, dusk/dawn). We provide the sequence IDs of these scenes in our codebase.

Table C.1: **Ablation study**.

| Setting | Scene Reconstruction | | Novel View Synthesis | | Scene Flow estimation |
|---|---|---|---|---|---|
| | Full Image PSNR↑ | Dynamic-Only PSNR↑ | Full Image PSNR↑ | Dynamic-Only PSNR↑ | Flow Acc$_5$(%) ↑ |
| (a) 4D-Only iNGP | 26.55 | 22.30 | 26.02 | 21.03 | - |
| (b) no flow | 26.92 | 23.82 | 26.33 | 23.81 | - |
| (c) no temporal aggregation | 26.95 | 23.90 | 26.60 | 23.98 | 4.53% |
| (d) freeze temporally displaced features before aggregation | 26.93 | 24.02 | 26.78 | 23.81 | 3.87% |
| (e) ours default | **27.21** | **24.41** | **26.93** | **24.07** | **89.74%** |

# C    ADDITIONAL RESULTS

## C.1    QUALITATIVE RESULTS

Figure C.1 shows the qualitative comparison to HyperNeRF (Park et al., 2021b) and D$^2$NeRF (Wu et al., 2022). `EmerNeRF` produces a more photorealistic temporal novel view and achieves better dynamic-static decomposition. In addition, our `EmerNeRF` can generate novel scene flows.

## C.2    ABLATION STUDIES

Table C.1 provides ablation studies to understand the impact of different components on novel view synthesis and scene flow estimation. The crucial role of the positional embedding removal method has been demonstrated in the main text. For the ablation purpose, models for all experiments here are trained for 8k iterations. Using a full 4D iNGP without the static field results in the worst results due to missing multi-view supervision. Incorporating hybrid representations consistently improves the performance (b-e). Without temporal aggregation step (c), or by freezing temporal feature gradients (d) (stop the gradients of $\mathbf{g}_d^{t-1}$ and $\mathbf{g}_d^{t+1}$ in Fig. 2), the flow estimation ability does not emerge (the last column). Applying all settings yields the best results.

## C.3    LIMITATIONS

Inline with other methods, `EmerNeRF` does not optimize camera poses and is prone to rolling shutter effects of cameras. Future work to address this issue can investigate joint optimization of pixel-wise camera poses and scene representations. In addition, the balance between geometry and rendering quality remains a trade-off and needs further study. Finally, we leave it for future work to investigate more complicated motion patterns.

## C.4    VISUALIZATIONS

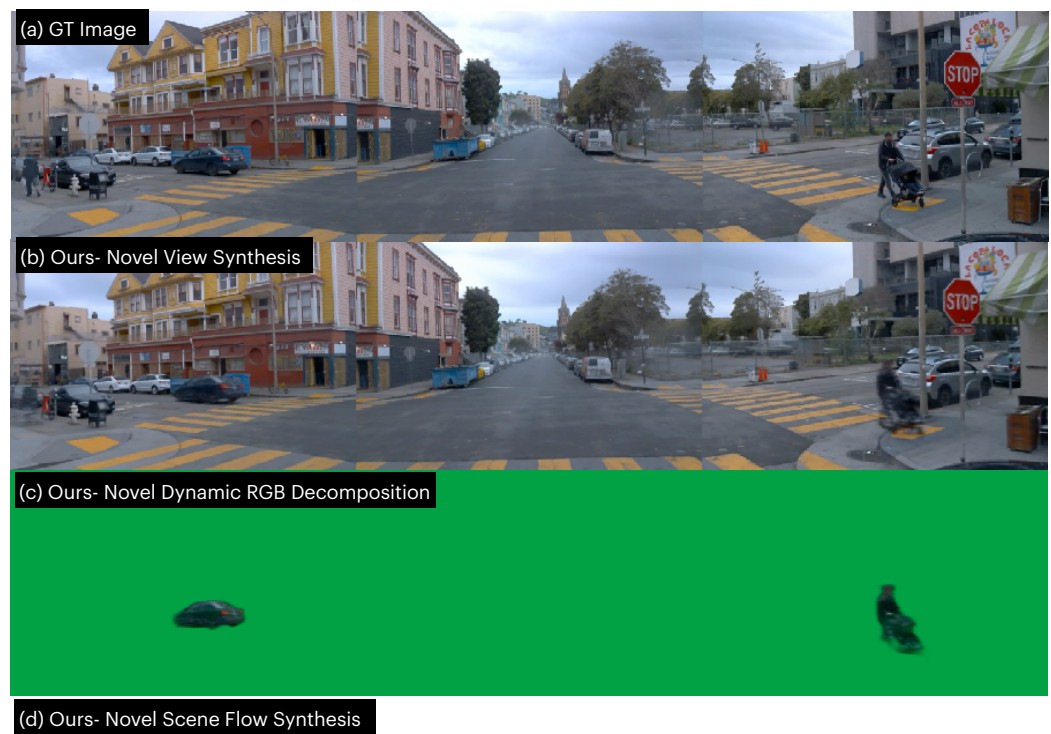

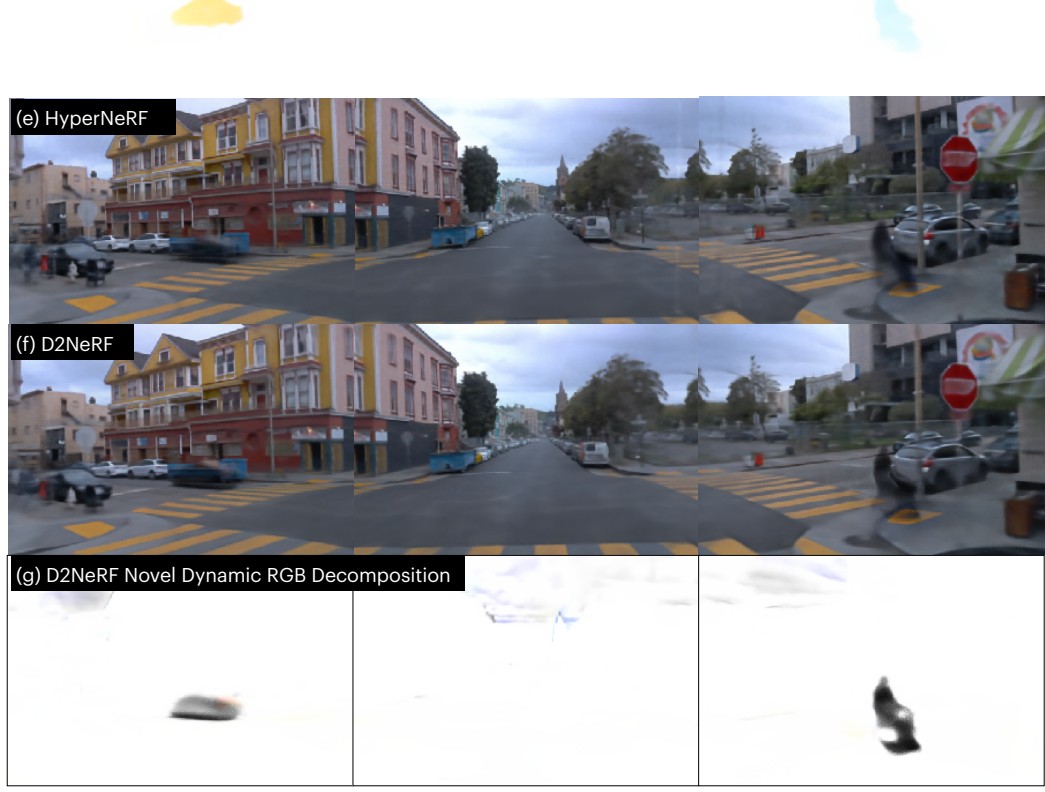

Figure C.1: Qualitative novel temporal view comparison.

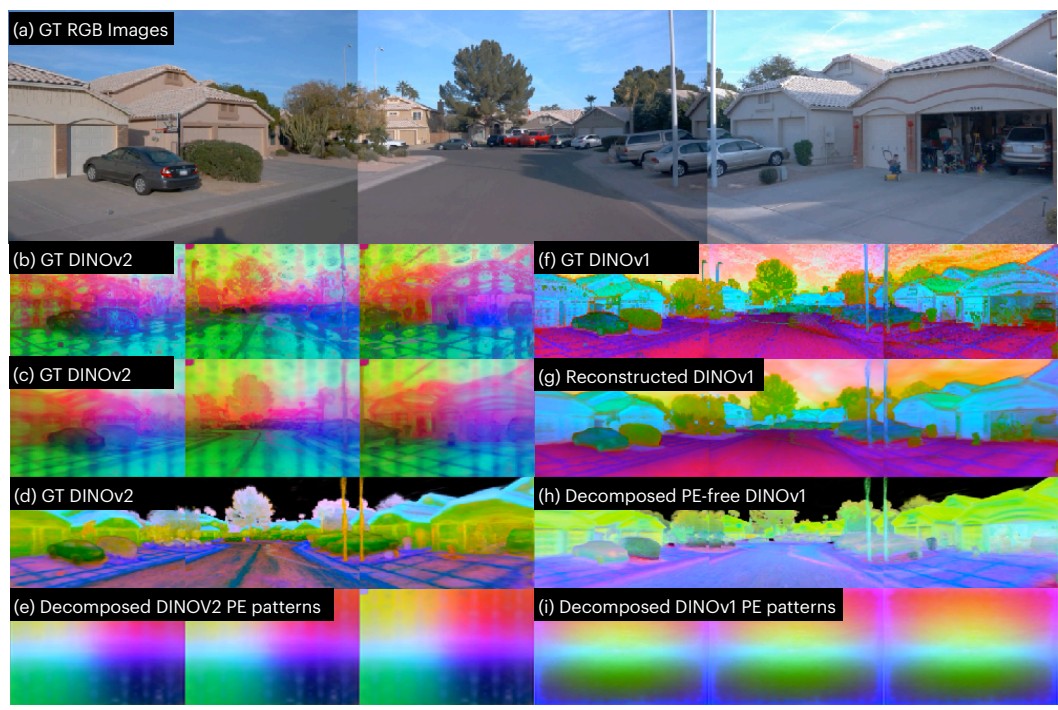

Figure C.2: Different positional embedding patterns in DINOv1 (Caron et al., 2021) and DINOv2 models (Oquab et al., 2023)

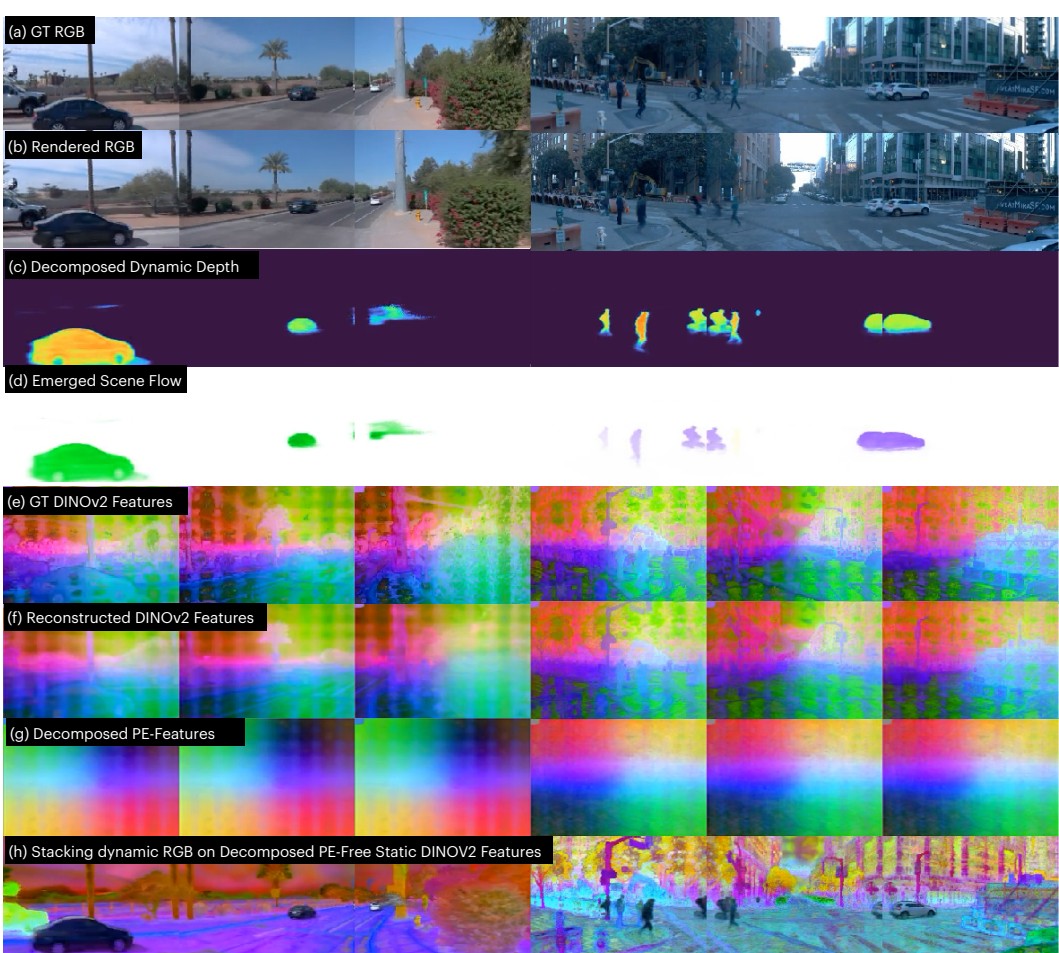

Figure C.3: Scene reconstruction visualizations of `EmerNeRF`. We show (a) GT RGB images, (b) reconstructed RGB images, (c) decomposed dynamic depth, (d) emerged scene flows, (e) GT DINOv2 features, (f) reconstructed DINOv2 features, and (g) decomposed PE patterns. We also stack colors of dynamic objects onto decomposed PE-free static DINOv2 features.

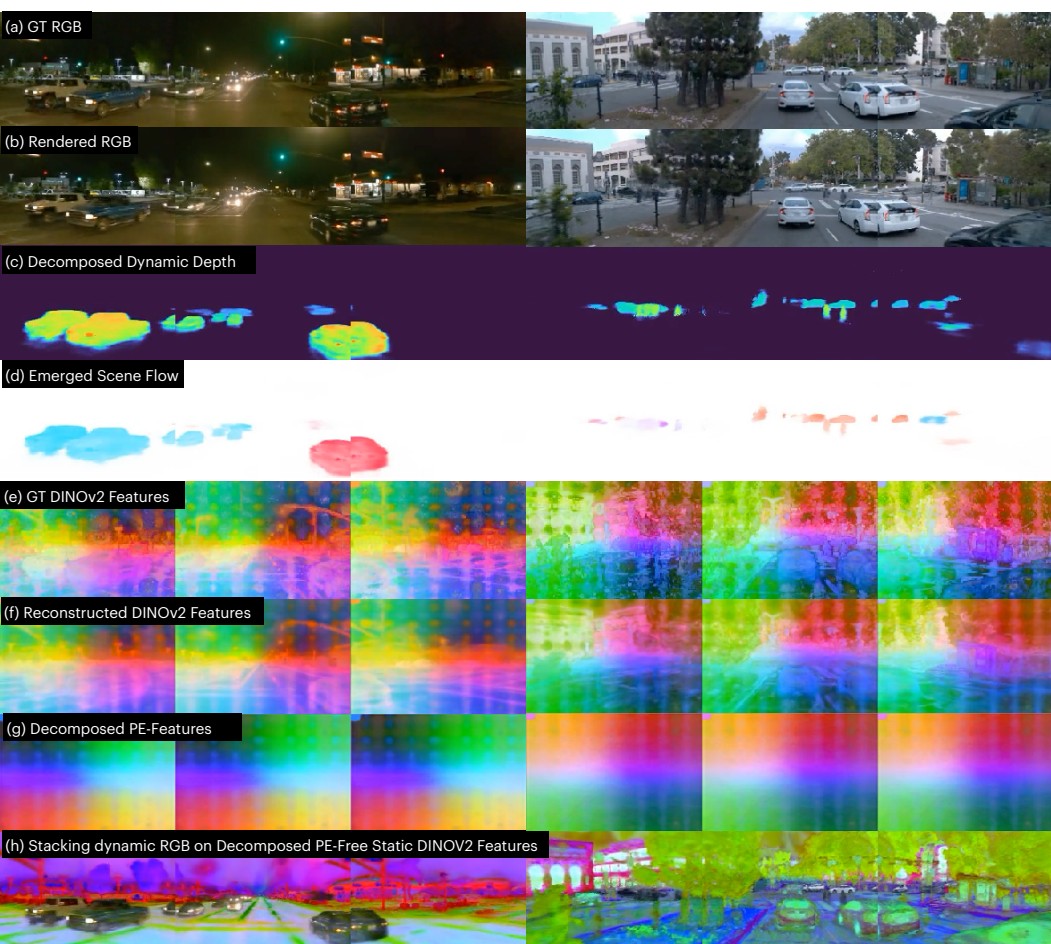

Figure C.4: Scene reconstruction visualizations of `EmerNeRF` under different lighting conditions. We show (a) GT RGB images, (b) reconstructed RGB images, (c) decomposed dynamic depth, (d) emerged scene flows, (e) GT DINOv2 features, (f) reconstructed DINOv2 features, and (g) decomposed PE patterns. We also stack colors of dynamic objects onto decomposed PE-free static DINOv2 features. `EmerNeRF` works well under dark environments (left) and discerns challenging scene flows in complex environments (right). Colors indicate scene flows' norms and directions.

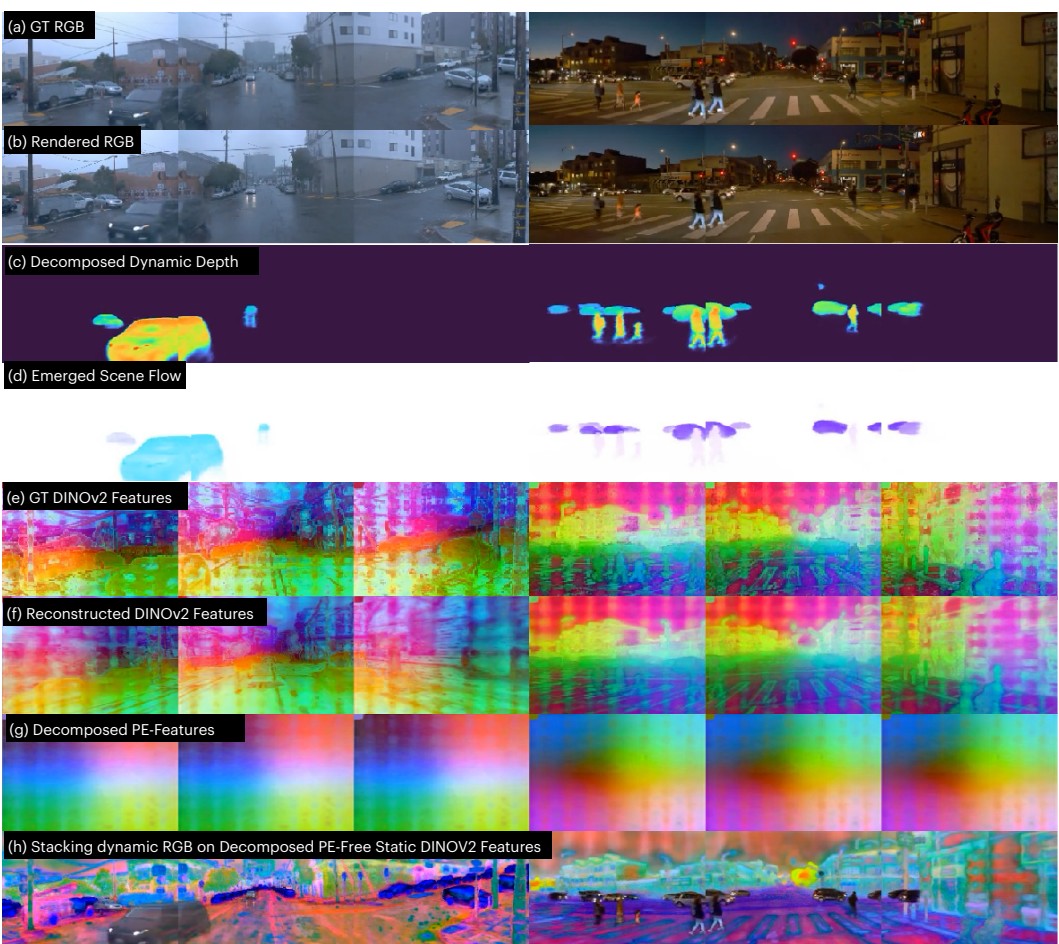

Figure C.5: Scene reconstruction visualizations of `EmerNeRF` under differet lighting and weather conditions. We show (a) GT RGB images, (b) reconstructed RGB images, (c) decomposed dynamic depth, (d) emerged scene flows, (e) GT DINOv2 features, (f) reconstructed DINOv2 features, and (g) decomposed PE patterns. We also stack colors of dynamic objects colors onto decomposed PE-free static DINOv2 features. `EmerNeRF` works well under gloomy environments (left) and discerns fine-grained speed information (right).