# OpenReview forum: "EmerNeRF: Emergent Spatial-Temporal Scene Decomposition via Self-Supervision"
_ICLR.cc/2024/Conference — ICLR 2024 poster_

### Official Review · Reviewer_fQX6 · 2023-10-30

**Soundness:** 2 fair
**Presentation:** 3 good
**Contribution:** 2 fair
**Rating:** 6
**Confidence:** 3

**Summary:**

This paper proposes EmerNeRF to decompose a scene into static and dynamic components by learning from videos using a self-supervised manner. The authors further propose to lift visual features from foundation models into the 4D space, by learning a shared PE feature map and PE-free volumetric feature fields. The authors also construct a new benchmark by subsampling video sequences from the Waymo Open Dataset. The proposed method achieves good results on several tasks, including scene reconstruction, view synthesis, and occupancy prediction.

**Strengths:**

1. The proposed method is intuitive, simple yet effective, achieving promising results on several tasks
2. I like the study of PE patterns in vision foundation models and the solution to it. The visualization in Figure 1 is also good.

**Weaknesses:**

**1. Method**
- First of all, the authors claim “no prior works in this field have explicitly modeled temporal correspondence for dynamic objects”, which is wrong. Some related works are missing here [A1-3].
- It seems to me that the formulation of Gao et al. [A2] is very similar to the proposed method. Except Gao et al. [A2] used: (1) MLP-based NeRF instead of hash grids; (2) separate RGB heads for static and dynamic branches; (3) Not using a sky branch. However, Gao et al. [A2] showed that their method would fail without the optical flow regularization. So I wonder what is the core reason that the proposed method can work while Gao et al. [A2] cannot. Is it due to the hash grid? Is it due to ground truth depth supervision (from lidar)? Is it due to the dataset being evaluated (multiview-view, vehicle motions are rigid and thus simpler)? Or is it due to the evaluation protocol (relatively easy to do view synthesis for frame interpolation)? While I prefer such a simple method, the reason that makes it work remains unclear to me.

**2. Experiments**
- For the results in Table 1 and Table 2, is visual feature distillation being used?
- Is it possible to show to number of parameters in Table 1?
- The comparisons with baselines in Table 1 also seems problematic. First, the authors project lidar points to image and use a L2 loss for depth regularization for HyperNeRF and D2NeRF, which is definitely worse than directly regularizing on lidar rays due to many reasons (e.g., projection error, possible occlusions). While the authors claim that neither of these two baselines support lidar ray sampling, it is actually not difficult to incorporate this functionality. An easier way is to use the same way to regularize depth in the proposed framework. Second, StreetSuRF has its own normal supervision and lidar ray preprocessing. Simply disabling them and use the same one as in the proposed method does not seem correct to me.
- While not necessary, would be good to see a comparison on the scene flow estimation task with a SOTA method [A4].


**Summary**

My main concern is that I don’t know what is the core reason that makes the proposed work. It would be great if the authors can conduct more extensive ablation analysis or conduct experiments on other datasets (especially those monocular ones used for dynamic view synthesis). Also, I would expect to see more reasonable comparisons with the baseline methods. The authors should also incorporate the missing literature and discussion accordingly.


**Reference**
- [A1] Wang et al. “Neural Trajectory Fields for Dynamic Novel View Synthesis”
- [A2] Gao et al. “​​Dynamic View Synthesis from Dynamic Monocular Video”. ICCV 2021
- [A3] Wang et al. “Flow supervision for Deformable NeRF”. ICCV 2023
- [A4] Li et al. “Fast Neural Scene Flow”. ICCV 2023

**Questions:**

Please see the weakness section.

---

> ### Author Response · Authors · 2023-11-22
> **Response to Reviewer fQX6 [1/4]**
>
> We thank the reviewer for the detailed review and insightful questions. These constructive comments provide valuable perspectives to enhance our work. Additionally, we are grateful for your recognition of the intuitive and effective nature of our proposed method, as well as your positive remarks on our study of PE patterns in vision foundation models.
>
> Before addressing the reviewer’s specific questions, we start by addressing your main concern stated in the summary section---the core motivation behind EmerNeRF. Our primary objective is to advance neural scene representations in the autonomous driving (AD) domain, an area that remains crucial yet underexplored. We view neural scene representations as more than just a tool for rendering; our goal is to develop a comprehensive scene representation that not only achieves high rendering quality but also effectively models geometry, motion, and semantics together. EmerNeRF is designed to fulfill these objectives. Therefore, we argue that the novelty and significance of our work should be evaluated within the broader context of its applicability and impact, particularly in the fields of autonomous driving and robotics. Our work lays an excellent foundation for future research, thanks to its open-sourced implementation, comprehensive task setups, and the inclusion of a diverse range of challenging driving sequences, all of which we believe will greatly facilitate and inspire subsequent studies in the field of autonomous driving and robotics.
>
> Below, we address the reviewer’s specific concerns:
>
> 1. **Missing related work**. We are grateful to the reviewer for highlighting additional relevant literature.  We have included a discussion of these related works in the revised version of our paper. Additionally, we plan to develop a detailed comparison section in the main text of our camera-ready paper to provide a more thorough analysis of how our work relates to these works.
> 2. **Distinction from Related Work in Flow Field**:
> ***Self-supervised flow***. EmerNeRF learns scene flows without external flow supervision. To our knowledge, all existing NeRF methods with flow fields default to using direct flow supervision. However, unlike color or depth, this supervision is impossible to capture via raw sensors such as cameras or LiDAR. Therefore, they have to rely on heuristic methods or pre-trained models to extract 2D optical flows. Yet, these flows often suffer from noise and frame-to-frame inconsistencies, necessitating post-processing steps to filter out inaccurate flows or the application of strong regularizations for successful training. EmerNeRF’s scene flow emerges from self-training, thereby eliminating the need for 2D optical flow supervision and  strong flow regularizations.

---

> ### Author Response · Authors · 2023-11-22
> **Response to Reviewer fQX6 [2/4]**
>
> 3. **Experiments**:
> - 1. **Feature distillation in Tables 1 & 2**: The feature head is disabled, as stated at the end of the “setup” paragraph in Section 4.1.
> - 2. **Number of parameters**. We have included the number of parameters and training times of different methods in Appendix A.3 for your reference. Specifically, StreetSurf employs a near-field hash encoder (32M) and a far-field encoder (16M). Our approach has
> a static hash encoder (30.55M parameters), with the optional inclusion of a dynamic hash encoder
> (10.49M parameters) and a flow encoder (9.70M parameters). It’s important to note that the number of parameters in both implicit and hybrid models does not directly correlate with their performance. For instance, as illustrated in Figure 2 of [1], an MLP with 438k parameters can achieve better image fitting than a hybrid model with a total of 33.6M parameters (10k from MLP and 33.6M from the grids). Please refer to the revised paper for more discussions.
>
> | Method         | # Parameters in MLPs | # Parameters in Grids | GPU Hours | # Iterations | Seconds / 1k Iterations | Batch Size |
> |----------------|----------------------|-----------------------|-----------|--------------|-------------------------|------------|
> | **Static:**    |                      |                       |           |              |                         |            |
> | StreetSurf     | 0.055M               | 32M + 16M             | 1.26 hrs  | 12,500       | 362.88s                 | 8192       |
> | Ours           | 0.065M               | 30.55M                | 0.58 hrs  | 25,000       | 83.52s                  | 8192       |
> | **Dynamic:**   |                      |                       |           |              |                         |            |
> | D2NeRF         | 3.03M                | 0                     | 9.68 hrs  | 100,000      | 348.48s                 | 4096       |
> | HyperNeRF      | 1.32M                | 0                     | 7.32 hrs  | 100,000      | 263.52s                 | 4096       |
> | Ours w/o flow  | 0.065M               | 30.55M + 10.49M       | 0.88 hrs  | 25,000       | 126.72s                 | 8192       |
> | Ours w/t flow  | 0.065M               | 30.55M + 10.49M + 9.7M| 2.03 hrs  | 25,000       | 292.32s                 | 8192       |
>
>
> - 3. **Lidar Projection**. We tested a variant of EmerNeRF that uses projected LiDAR points onto the image plane for depth supervision. This modification only marginally affects performance, indicating it is not a key factor that makes our method outperform other baselines. The results are reported in Table C.2. in Appendix C. We also provide the numbers here for reference:
>
> | Method | Scene Reconstruction: Full Image | Scene Reconstruction: Dynamic-Only | Novel View Synthesis: Full Image | Novel View Synthesis: Dynamic-Only |
> | ------ | -------------------------------- | ---------------------------------- | ------------------------------- | ---------------------------------- |
> |        | PSNR↑ / SSIM↑                    | PSNR↑ / SSIM↑                      | PSNR↑ / SSIM↑                   | PSNR↑ / SSIM↑                      |
> | use projected depth     | 28.874 / 0.8097                 | 26.200 / 0.7275                   | 27.532 / 0.7884                 |  24.321 / 0.6642     |
> | default | 28.873/ 0.8096                  | 26.223 / 0.7276                  | 27.547 /  0.7885                  | 24.323 /  0.6655
>
> - 4. **StreetSurf Results**. The reported PSNR in the StreetSurf paper for the same dataset split is 26.66, achieved with normal supervision and their processed LiDAR rays. Our re-run of their code under the same setting yielded a PSNR of 26.44. Our initial submission reported a PSNR of 26.15 when trained without normals and using our processed data. In contrast, our method achieves a PSNR of 28.54. We believe our reproduction of their results is faithful. The number in our paper has been updated to reflect the 26.66 PSNR as reported in the StreetSurf paper.
>
> - 5. **Comparison to FastNSFF**: We thank the reviewer for pointing out the recent baseline as a potential comparison for our work. However, due to the limited timeframe and an intensive CVPR effort during the rebuttal period, we were unfortunately unable to complete the comparison experiment with FastNSFF by the rebuttal deadline. We recognize the importance of this comparison and are committed to including it in the camera-ready version of our paper. We appreciate your understanding.
>
> Again, We sincerely thank the reviewer for initiating these insightful discussions to further strengthen EmerNeRF. We apologize for the delayed response due to the CVPR deadline. We welcome any further comments or suggestions and commit to addressing them thoroughly in the camera-ready version of our paper.
>
> [1] Müller, Thomas, et al. "Instant neural graphics primitives with a multiresolution hash encoding." ACM Transactions on Graphics (ToG) 41.4 (2022): 1-15.

---

> ### Author Response · Authors · 2023-11-22
> **Responses to Reviewer fQX6 [3/4]**
>
> 4. **What is the key factor to make EmerNeRF excel at flow estimation?**
> TL;DR: The key to our success lies in proper depth supervision and temporal feature aggregation.
> - **1. Proper Depth supervision**: Consistent with other NeRF applications in the AD domain, EmerNeRF relies on LiDAR data for accurate scene reconstruction. However, we argue that depth signals, commonly obtained through LiDAR sensors, are readily available in the AD context and are integral to various tasks. Given the widespread availability of LiDAR sensors and the absence of a direct "flow" sensor equivalent, the challenge of learning flows from camera images and LiDAR data emerges as a significant area of interest. We also want to highlight the novelty of our approach in this regard.
>
> Additionally, we provide quantitative results to identify the importance of different forms of depth losses, which are presented in Appendix C. Please feel free to refer to there for detailed settings. We include the results to provide a holistic preview. Specifically, (b) Using an L2 depth loss results in reasonable flow estimations  (i.e., achieving satisfactory Acc5 and Acc10 results), but these
> estimates come with a large angle error. (c) Applying a line-of-sight loss yields much better results in terms of angle errors. The line-of-sight loss encourages a concentrated density distribution along a ray towards its termination point—a peaky distribution. This strategy effectively resolves the ambiguity associated with accumulating numerous noisy flows in the 3D scene, as it focuses on the core contributors of these flows: the dynamic objects. (a) Combining these two depth losses provides the optimal performance.
>
> | Settings                                                     | EPE3D (m) ↓ | Acc5(%) ↑ | Acc10(%) ↑ | θ (rad) ↓ |
> |--------------------------------------------------------------|-------------|-----------|------------|-----------|
> | Combined L2 and line-of-sight loss for depth; feature-ensemble
> | (a) Ours                                             | **0.0130**      | 0.9272    | **0.9599**     | **0.5175**    |
> | Depth supervision:                                          |             |           |            |           |
> | (b) L2 loss only                                             | 0.0367      | 0.9077    | 0.9211     | 1.1207    |
> | (c) Line-of-sight-loss only                                  | 0.0136      | **0.9296**    | 0.9574     | 0.5996    |

---

> ### Author Response · Authors · 2023-11-23
> **Rebuttal follow up**
>
> Dear reviewer,
>
> We greatly appreciate the time and effort you have invested in reviewing our manuscript. In response, we have carefully considered your feedback and have addressed the concerns you raised in our rebuttal.
>
> We understand how busy you must be, especially as we approach the end of the discussion period. We look forward to and would be grateful for any final comments you could provide, and the rating score you could raise. Your feedback is crucial to us.

---

> ### Author Response · Authors · 2023-11-23
> **[4/4] Continued of What is the key factor to make EmerNeRF excel at flow estimation?**
>
> - **Temporal Feature Aggregation for Robust Learning:**
> In existing NeRF-based methods, flow fields are typically employed in a three-step process (which we refer to as “Color-warping” in the following text):
> 1. Temporal Warping: Sampling points are warped to their positions at the previous and/or next timestep to query their corresponding density and colors from nearby frames.
> 2. Accumulation: These densities and colors are then accumulated along corresponding rays in the current time step. Note that for each ray, we perform three individual integrations for density and colors from the past, the current, and the next timestep. This construction results in three distinct per-ray colors.
> 3. RGB Loss Computation: The RGB loss is calculated for each of these three colors.
>
> However, the warped points are occluded in the previous or next timestep, making the RGB loss computation for these points invalid. As a solution, NSFF [2] proposed a loss weighting strategy to alleviate this issue. Nonetheless, this approach, referred to as the “warping-based method,” is computationally expensive as it requires querying the RGB network three times and rendering three distinct sets of rays.
>
> Our approach innovates by aggregating the warped features from both the previous and next timestep. We only query the RGB network with aggregated features and render a single color per ray. This method effectively bypasses the occlusion issues associated with forward- and backward-warped renderings by focusing solely on the current timestep, while leveraging information from nearby frames. Additionally, by decoding the aggregated features through a color MLP, we effectively mitigate potential artifacts introduced by warping, leading to more robust rendering.
>
> To validate this hypothesis, we introduce two variations of our EmerNeRF:
>
> - Variation 1: Renders forward- and backward-warped rays separately, applying reconstruction losses to each (resulting in three colors per ray in the current timestep), as discussed above. We call this method “**Color-warping**”.
> - Variation 2: Implements color aggregation instead of feature aggregation. This involves predicting forward- and backward-warped per-point colors, aggregating them by taking an average of per-3D location colors from the previous, the current, and the next timestep before taking integration along each ray, and rendering the final output (resulting in one color per ray). We call this method “**Color-ensemble**”. This is to study if the color MLP helps to mitigate feature noise.
>
> Our results indicate that the default “Feature-ensemble” (our method) outperforms others, with the “Color-ensemble” showing the least effective results. The “Color-warping” variant also demonstrated less accurate flow estimations compared to our default method while taking longer training time due to its extra network queryings and color renderings. In addition, all these methods demonstrate promising flow estimation ability. This analysis underscores the potential of utilizing neural fields for flow estimation, especially when depth signals are available, eliminating the need for heuristic methods or pre-trained models to provide optical flow labels. We believe this observation will inspire a broader range of follow-up studies.
>
>
>
> | Settings                                                     | EPE3D (m) ↓ | Acc5(%) ↑ | Acc10(%) ↑ | θ (rad) ↓ |
> |--------------------------------------------------------------|-------------|-----------|------------|-----------|
> | (a) Feature-ensemble   (ours)          | **0.0130**      | 0.9272    | **0.9599**     | **0.5175**    |
> | (b) Color-warping                                      | 0.0190      | 0.9181    | 0.9370     | 0.6868    |
> | (c) Color-ensemble                                      | 0.0414      | 0.8929    | 0.9065     | 1.5708    |
>
> [2] Li, Zhengqi, et al. "Neural scene flow fields for space-time view synthesis of dynamic scenes." Proceedings of the IEEE/CVF Conference on Computer Vision and Pattern Recognition. 2021.

---

### Official Review · Reviewer_hxFT · 2023-10-30

**Soundness:** 3 good
**Presentation:** 4 excellent
**Contribution:** 3 good
**Rating:** 8
**Confidence:** 3

**Summary:**

This paper presents a NeRF based method for learning scene representation. By decomposing the loss function into two separate terms - one handling static elements of the scene and the other dynamic elements the model is able to separate the static and dynamic components with no extra supervision. The model also estimates scene flow in the process as part of the regularization. Finally the method is also applied to "lifting" 2D self-supervised representations (e.g. DINO) into 4D space by a combination of readout heads and a learned positional embedding which rectifies some of the issues associated with these SSL representations.

**Strengths:**

The paper presents a nice combination of ideas that have been floating around for a while, leveraging the strengths of different approaches in an appealing and relatively elegant way.
The paper is well presented, well executed and results are impressive all in all (but see below) - this is a good paper.

**Weaknesses:**

I think there are a couple of weaknesses that may require addressing

* Ablation analysis is lacking - table C1 addresses some of the of the modeling decisions but there are more aspects I would have loved to see analyzed.
* Applicability - I may be wrong, but I feel the use of driving data together with a NeRF based method (that is, need to train on each scene separately) is a bit odd - usually in this use case one would want an online inference model (e.g. an encoder) which can infer elements and structure quickly as the car/robot drives around the scene. On the other hand I feel this is not widely applicable to other types of data (say free form natural scenes with lots of unstructured movement).
* There is a lot of focus on DINO/v2 features and their related issues - I am wondering if other SSL methods suffer from similar PE issues (specifically ones that handle motion such VideoMAE / Siamese MAE etc.)

**Questions:**

In relation to the above:

* How much dependance is there on LiDAR data? would this method work without direct depth supervision signals?
* It's not written explicitly in the paper - does the data include ground truth camera parameters? (extrinsic and intrinsic, I imagine the answer is yes)
* The dynamics regularization term is a bit simplistic as it is a simple minimization of total density for dynamic elements - have you tried other regularization methods? say assuming sparsity and so on?

---

> ### Author Response · Authors · 2023-11-22
> **Response to Reviewer hxFT**
>
> Dear reviewer, we sincerely appreciate the time and effort you dedicated to reviewing our paper. Your thoughtful and constructive feedback has been invaluable in highlighting the strengths of our work. Your insightful reviews and acknowledgment of our contributions inspire us to continue refining our work in this field. We address your concerns below:
>
> **Ablation analysis**. We appreciate your emphasis on the importance of a detailed ablation analysis. In response, we have expanded this section in the appendix to include a more extensive range of studies. These studies are conducted across 5 sequences randomly selected from the Dynamic32 split, specifically for ablation purposes. The additional ablations include a variety of configurations, such as head ablation, depth loss variations, feature field ablation, and flow ablation studies. These extended analyses offer more insights into the impact of each component within our model. Due to the extensiveness of the results, we refer the reviewer to Appendix C in the revised paper for details.
>
>
> **Applicability of NeRF methods for driving**. We appreciate you initiating this great discussion.  The major applications this paper considers are auto-labeling and sensor simulation for autonomous driving. E.g., one can combine this sensor simulation with traffic modeling to benchmark a planning algorithm with such real-world captured data, instead of using a graphics engine to simulate scenes. In addition, EmerNeRF can enable large-scale reconstruction from raw sensor data. We also agree with reviewers that investigating a generalizable pipeline for online inference will be valuable. We are actively working on a large world model that can infer scene structures and semantics from online observations with an encoder architecture. We see EmerNeRF as a first step towards this goal as it provides: 1) a flexible scene representation that considers semantics, geometry, and motion, and 2) pseudo labels and supervision for this world model. We are excited about this direction and would like to take your inputs!
>
> **Discussion regarding ViT features**.
> This is indeed another great and insightful question, which closely aligns with our current research focus. In fact, we have an immediate follow-up work under review of CVPR where we discovered that these artifacts exist in nearly all Vision Transformers. We traced their origins to the positional embedding added to the input tokens and proposed a method to address this issue across all ViTs. We are also curious about whether 3D video ViTs (that explicitly deal with video data, like VideoMAE / Siamese MAE) exhibit a similar issue. However, our study on Video ViTs is not yet complete, as how to lift video features to 4D remains unexplored within the NeRF community. We are actively pursuing this line of research and are eager to share our findings on our anonymous website as soon as new updates are available. Although these updates might be beyond the scope of this rebuttal, we invite the reviewer and interested readers to check our anonymous website for new visualizations and findings, which we plan to post soon. We seriously consider your question on 3D video ViTs and are more than glad to explore this direction further!
>
> **Addressing specific questions:**
> 1. **Dependence on LiDAR data**: Our method, in line with other NeRF approaches for autonomous driving (AD), relies on LiDAR data to reconstruct high-quality geometry.
> 2. **Ground truth camera parameters**. Yes, we do use the ground truth camera parameters that are readily available in all AD datasets.
>
> Despite these two mentioned aspects are beyond the scope of this paper, we acknowledge their importance for both research and real-world applications. Recognizing the limitations they may pose, we aim to improve these aspects in future work by exploring ways to reduce or eliminate the dependence on LiDAR and ground truth camera parameters.
>
> We believe our work will provide an excellent foundation for future research endeavors aimed at tackling these challenges, thanks to the open-sourced nature of our implementation, the comprehensive nature of our task setups, and the inclusion of a diverse range of challenging driving sequences. We anticipate that these elements will greatly facilitate and inspire subsequent studies in this field.
>
> 3. **Other regularization terms**: At the beginning of this project, we experimented with several sparsity regularization terms and converged to the current form. We commit to including this ablation study in the final version.
>
> We thank the reviewer again for the valuable feedback and hope that these additional details and clarifications address your concerns. We apologize for our late response and understand the rebuttal period is closing soon, but please let us know if you have any further questions. We commit to incorporating as many enhancements as possible in the camera-ready version of the paper.

---

> > ### Comment · Reviewer_hxFT · 2023-11-23
> > **Thank you for your detailed response!**
> >
> > I appreciate the time taken to answer my concerns! I definitely think this has made an already quite good paper, better!
> > I am keeping my score as it is (this is a good paper!) and I am hoping this will be accepted to the conference.

---

> > > ### Author Response · Authors · 2023-11-23
> > > **Thank you for your feedback**
> > >
> > > Dear reviewer, thank you so much for your valuable feedback and recognition of our paper! We will greatly appreciate an increased score if you find it becomes better---we strive to achieve the strongest possible submission we could have :-) We are committed to addressing any further concerns!

---

### Official Review · Reviewer_BkKZ · 2023-10-31

**Soundness:** 3 good
**Presentation:** 3 good
**Contribution:** 3 good
**Rating:** 8
**Confidence:** 4

**Summary:**

This paper presents a neural field approach that can perform static-dynamic decomposition and scene flow estimation in challenging autonomous driving scenarios. A hybrid 4D scene representation consisting of static, dynamic, and flow fields is adopted and jointly trained with the goal of appearance and feature reconstruction. Experimentation reveals state-of-the-art performance in novel view synthesis and scene flow estimation.

**Strengths:**

This paper proposes an effective way of jointly modeling the static and dynamic scenes in the setting of autonomous driving. The method is technically sound, from the high-level idea of using hybrid representation for static and dynamic scenes and optimize them under the goal of appearance reconstruction to the details of carefully modeling sky and shadows in the framework.

Even though there is no direct flow supervision or well-adopted supervision such as flow based warping, it's quite novel to see the scene flow estimation "emerges" from the temporal aggregation of features for scene reconstruction.

The experimentation is thorough and the numerical improvements over baselines are obvious.

The writing and presentation of this paper are also pretty good and easy-to-follow.

**Weaknesses:**

It's good to see more visualization of the model output in the appendix. But it would also be good to include more qualitative comparisons against the baselines in addition to the quantitative results. Also, adding error maps would be more intuitive to highlight the difference.

There also seems to miss the runtime analysis and comparison. How long does this approach take for sensor simulation in training and test time, and how does it compare to the existing approaches? That would be an important piece of information.

**Questions:**

The experiment setup omit every 10th frame, resulting in 10% novel views for evaluation. It would be better to include ablation study on the sparsity of the sampling and how the method degrade with fewer training frames.

---

> ### Author Response · Authors · 2023-11-22
> **Response to Reviewer BkKZ**
>
> Dear reviewer, we sincerely appreciate the time and effort you dedicated to reviewing our paper. We truly appreciate the positive reception of our paper and are keen to address the specific points the reviewer has highlighted. The feedback is invaluable for refining our work, and we look forward to discussing these aspects in more detail. Below we provide our preliminary responses:
>
>
> 1. **Visualizations**: We appreciate the suggestion regarding visualizations. In response, we have added more visual content to the appendix to enhance the qualitative comparison against the baselines. We promise to add more visualizations and their corresponding error maps as soon as possible. Additionally, we will update our anonymous website with visualizations that compare our methods with the baseline approaches. We greatly appreciate further suggestions and commit to incorporating as many enhancements as possible in the camera-ready version of the paper.
>
>
> 2. **Training and testing time**. Regarding the runtime analysis, we initially discussed training and testing times in section A.1 of the supplementary materials. For clarity, here's a detailed runtime comparison with previous works.
>
> | Method         | # Parameters in MLPs | # Parameters in Grids | GPU Hours | # Iterations | Seconds / 1k Iterations | Batch Size |
> |----------------|----------------------|-----------------------|-----------|--------------|-------------------------|------------|
> | **Static:**    |                      |                       |           |              |                         |            |
> | StreetSurf     | 0.055M               | 32M + 16M             | 1.26 hrs  | 12,500       | 362.88s                 | 8192       |
> | Ours           | 0.065M               | 30.55M                | 0.58 hrs  | 25,000       | 83.52s                  | 8192       |
> | **Dynamic:**   |                      |                       |           |              |                         |            |
> | D2NeRF         | 3.03M                | 0                     | 9.68 hrs  | 100,000      | 348.48s                 | 4096       |
> | HyperNeRF      | 1.32M                | 0                     | 7.32 hrs  | 100,000      | 263.52s                 | 4096       |
> | Ours w/o flow  | 0.065M               | 30.55M + 10.49M       | 0.88 hrs  | 25,000       | 126.72s                 | 8192       |
> | Ours w/t flow  | 0.065M               | 30.55M + 10.49M + 9.7M| 2.03 hrs  | 25,000       | 292.32s                 | 8192       |
>
> Hybrid methods, such as our proposed EmerNeRF and StreetSurf, can be trained in a relatively short duration of 30 minutes to 2 hours. This efficiency comes at the expense of utilizing explicit grids for encodings. In contrast, MLP-based models like D2NeRF and HyperNeRF have fewer parameters but require longer training times. It’s important to note that the number of parameters in both implicit and hybrid models does not directly correlate with their performance. For instance, as illustrated in Figure 2 of Muller et al. (2022), an MLP with 438k parameters can achieve better image fitting than a hybrid model with a total of 33.6M parameters (10k from MLP and 33.6M from the grids).
>
> **Ablation on the sparsity of the sampling**. We are thankful for the suggestion of conducting an ablation study on the sparsity of sampling. We promise to include this analysis in the camera-ready version, as running full experiments for all 32 scenes under different sampling ratios requires non-trivial computational resources.
>
> We hope these additional details and clarifications further strengthen our submission and address the concerns. We thank the reviewer again for the valuable feedback and hope that these additional details and clarifications address your concerns effectively. We apologize for our late response due to the limited timeframe and an intensive CVPR effort during the rebuttal period. Given that the rebuttal period is closing soon, feel free to let us know if you have any further questions. We commit to incorporating as many enhancements as possible in the camera-ready version of the paper. If our responses successfully address your concerns, we will greatly appreciate an increased score. Thank you again for all the support to make our submission stronger.

---

### Official Review · Reviewer_qWy5 · 2023-11-01

**Soundness:** 2 fair
**Presentation:** 3 good
**Contribution:** 3 good
**Rating:** 6
**Confidence:** 4

**Summary:**

The paper introduces EmerNeRF, an approach for learning spatial-temporal representations of dynamic scenes. EmerNeRF decomposes scenes into static and dynamic fields, with an additional scene flow field modeling the movement of objects across time. The dynamic feature is computed as a weighted sum of features from nearby timesteps, where the sampling operation is determined by the self-supervised scene flow field. Additionally, the paper proposes a method to generate positional encoding (PE) free features from a pretrained feature encoder by leveraging the time-consistent property of PE features. The empirical result indicates the proposed method can achieve better novel view synthesis, flow estimation, and few-shot semantic prediction results compared to the baselines.

**Strengths:**

- The proposed dataset holds potential value for research in dynamic scene reconstruction field
- EmerNeRF demonstrates better reconstruction quality in driving scene dataset compared to the baselines
- The obtained scene flow exhibits high accuracy compared to baseline method

**Weaknesses:**

- Novelty. Many design components of EMerNeRF have been proposed in previous work, including separated static and dynamic fields [2][3], sky head [1], shadow head [2], flow field [4]. The paper lacks a detailed discussion highlighting the differences compared to these previous works.
- Generalizability. The proposed method lacks verification in existing dynamic scene datasets used in baselines such as Nerfies [5], HyperNeRF [6]. Those datasets contain more complex deformations and a significant proportion of dynamic components. It is necessary to evaluate the robustness and understand the limitations of the proposed design modules, including dynamic density regularization and self-supervised flow field.
- Ambiguity of equation 6. The meaning and formulation of the expectation of the dynamic density remain unclear.
- Baseline. The quantitative evaluation lacks a more recent baseline [7].

[1] Konstantinos Rematas, Andrew Liu, Pratul P Srinivasan, Jonathan T Barron, Andrea Tagliasacchi, Thomas Funkhouser, and Vittorio Ferrari. Urban radiance fields. In *Proceedings of the IEEE/CVF Conference on Computer Vision and Pattern Recognition*, pp. 12932–12942, 2022

[2] Tianhao Wu, Fangcheng Zhong, Andrea Tagliasacchi, Forrester Cole, and Cengiz Oztireli. Dˆ2nerf: Self-supervised decoupling of dynamic and static objects from a monocular video. *Advances in Neural Information Processing Systems*, 35:32653–32666, 2022.

[3] Martin-Brualla, Ricardo, Noha Radwan, Mehdi SM Sajjadi, Jonathan T. Barron, Alexey Dosovitskiy, and Daniel Duckworth. "Nerf in the wild: Neural radiance fields for unconstrained photo collections." In *Proceedings of the IEEE/CVF Conference on Computer Vision and Pattern Recognition*, pp. 7210-7219. 2021.

[4] Du, Yilun, Yinan Zhang, Hong-Xing Yu, Joshua B. Tenenbaum, and Jiajun Wu. "Neural radiance flow for 4d view synthesis and video processing." In *2021 IEEE/CVF International Conference on Computer Vision (ICCV)*, pp. 14304-14314. IEEE Computer Society, 2021.

[5] Keunhong Park, Utkarsh Sinha, Jonathan T Barron, Sofien Bouaziz, Dan B Goldman, Steven M Seitz, and Ricardo Martin-Brualla. Nerfies: Deformable neural radiance fields. In *Proceedings of the IEEE/CVF International Conference on Computer Vision*, pp. 5865–5874, 2021a.

[6] Keunhong Park, Utkarsh Sinha, Peter Hedman, Jonathan T Barron, Sofien Bouaziz, Dan B Goldman, Ricardo Martin-Brualla, and Steven M Seitz. Hypernerf: A higher-dimensional representation for topologically varying neural radiance fields. *arXiv preprint arXiv:2106.13228*, 2021b.

[7] Cao, Ang, and Justin Johnson. "Hexplane: A fast representation for dynamic scenes." In *Proceedings of the IEEE/CVF Conference on Computer Vision and Pattern Recognition*, pp. 130-141. 2023.

**Questions:**

- What’s the novel components that EmerNeRF propses?
- How does EmerNeRF perform in more general dynamic scenes?
- How does EmerNeRF compare to recent newer baseline?

---

> ### Author Response · Authors · 2023-11-22
> **Response to Reviewer qWy5 [1/2]**
>
> We extend our sincere thanks to the reviewer for their constructive feedback and insightful questions. We are encouraged by the reviewer's recognition of the value our method brings to novel view synthesis, flow estimation, and few-shot semantic prediction, as well as the significance of our dataset. Below, we address the specific questions raised in the review.
>
> ### **Novelty and Contribution**
> EmerNeRF’s major objective is to advance neural scene representations in the autonomous driving (AD) domain, a critical area that is less explored. When evaluating the novelty and, more importantly, the contribution of our work, we hope reviewers can take this application area into account. At a high level, EmerNeRF tackles extremely challenging problems that previous methods like BlockNeRF and UniSim (both CVPR oral/spotlight works) can’t solve, i.e., self-supervised scene decomposition. Here, we summarize and emphasize our contributions:
> 1. **Significance for AD and robotics:** EmerNeRF stands as a significant milestone in neural sensor simulators and auto-labeling for driving. The ultimate goal of EmerNeRF is to serve autonomous driving, a paramount real-world application. Typical challenges in integrating NeRF into driving include achieving high rendering quality, modeling dynamic objects, and enabling novel representations for scene semantic understanding/auto-labeling. EmerNeRF is the first and, so far, the only framework to comprehensively address these issues via a unified pipeline. Previous efforts in the AD domain, such as the well-known BlockNeRF, NSG, UniSim, and StreetSurf, achieve high rendering quality but heavily rely on ground truth annotations for segmenting and tracking objects. In contrast, EmerNeRF achieves self-supervised dynamic object disentanglement and motion estimation and provides robust semantics for scene understanding. This holistic capability positions EmerNeRF as a pioneering contribution to neural fields for AD.
>
> 2. **Scene Flow Estimation**: EmerNeRF learns scene flows without external flow supervision—an ability that no prior NeRF-based work has shown. To our knowledge, all existing NeRF methods with flow fields default to using direct flow supervision. However, unlike color or depth, this supervision is impossible to capture via raw sensors such as cameras or LiDAR. Therefore, they have to rely on heuristic methods or pre-trained models to extract optical flows, which are often noisy and inconsistent across frames, subsequently requiring post-processing steps and strong regularizations to ensure successful training. In striking contrast, our method learns scene flows purely from raw sensor data via the proposed temporal aggregation method. None of the prior works has revealed this emerging ability to estimate flow using neural fields; EmerNeRF is the first to showcase this potential, enlightening a new path to eliminate the need for flow labels. Additionally, scene flow estimation is a long-standing problem in autonomous driving, and EmerNeRF immediately establishes itself as state-of-the-art for this task. We believe our work will resonate with broader communities, including but not limited to dynamic NeRF and autonomous driving/robotic communities.
> 3. **Enhancing Feature Fields**: EmerNeRF is the first, if not the only, to identify and address the overlooked issue of 2D positional embedding artifacts in lifting ViT features via neural fields. There is increasing interest in lifting ViT features via neural fields to ground semantics in 3D, as seen in works such as LeRF and Distilled Feature Field. However, our comprehensive visualizations and quantitative results indicate that ignoring PE artifacts in ViT models can lead to significant performance drops. Our proposed method is not only simple but also universally applicable, potentially benefiting a wide range of feature field works within and beyond the NeRF community.
>
> In conclusion, EmerNeRF introduces significant advancements and provides insights into “NeRF as dynamic scene representations for AV and robotics”, demonstrating the potential that extends beyond its current applications.

---

> ### Author Response · Authors · 2023-11-22
> **Response to Reviewer qWy5 [2/2]**
>
> **Difference from [1-4].** In addition, we acknowledge the contribution of [1-4] and included all these works in our initial submission. We use similar sky head [1] and shadow head [2] in this paper. However, our design of static, dynamic, and flow field is novel in the sense that no prior work successfully learns a decomposition of these three fields completely through self-supervision (D2NeRF only learns static-dynamic decomposition without flow modeling). Moreover, our method provides a simple yet effective pipeline to integrate all these individual contributions to work synergistically.
>
> **New dataset and Baselines.** We acknowledge the concern regarding the generalizability of EmerNeRF. We consider the application of our framework to other dynamic-NeRF datasets as an orthogonal contribution, which lies beyond the scope of this particular study.
> Yet, we still would like to include their results.
> Unfortunately, due to the limited timeframe and an intensive CVPR effort during the rebuttal period, we are unable to finish all suggested experiments on a new dataset and re-implementation of other methods, we are committed to including these results in the anonymous website in the coming weeks and add them into the camera-ready version of our paper. We hope reviewers can consider these results on the website during the discussion period should they be of interest. We also welcome further suggestions and feedback to make this paper stronger. We sincerely thank the reviewer’s efforts in helping get this submission into a better shape! We also thank the reviewer for pointing out the recent baseline as a potential comparison for our work. We have included a discussion of this paper in the related work section, and we commit to incorporating this baseline in the camera-ready version of our paper.
>
>
> **Ambiguity in Equation 6**, We apologize for the confusion caused by the presentation of Equation 6. This equation represents a regularization term designed to minimize the expectation of dynamic density, i.e., we encourage dynamic density to be as small as possible and prompt the dynamic field to produce density values only as needed. We have revised the paper to include a clearer description and formulation of this aspect.

---

> ### Author Response · Authors · 2023-11-22
> **Rebuttal follow up**
>
> Dear reviewer,
>
> We greatly appreciate the time and effort you have invested in reviewing our manuscript. In response, we have carefully considered your feedback and hopefully have addressed the concerns you raised in our rebuttal.
>
> We understand how busy you must be, especially as we approach the end of the discussion period. Still, we look forward to and would be grateful for any final comments you could provide, and the rating score you could raise. Your feedback is crucial to us.

---

### Meta-Review · Area_Chair_fPcY · 2023-12-06

**Metareview:**

This paper proposes a self-supervised approach for learning spatial temporal features for driving scenes which can be applied to autonomous vehicles. Overall, the paper is fairly well written, the experiment is comprehensive and the approach achieved good empirical performance. The main weaknesses that the reviewer brought up are novelty and clarifications regarding the relation to prior works. The AC believes that the authors' rebuttal has addressed these concerns.

**Justification For Why Not Higher Score:**

The contribution of the paper is sufficient for publication but may not be of interest to a general audience.

**Justification For Why Not Lower Score:**

The AC believes that the concerns of the two borderline rejects were address by the authors' rebuttal.

---

### Decision · Program_Chairs · 2024-01-16

Accept (poster)